# L2T-DLN: Learning to Teach with Dynamic Loss Network

**Zhaoyang Hai** [1,*], **Liyuan Pan** [1,2,*,†], **Xiabi Liu** [1,*,†], **Zhengzheng Liu**[1], **Mirna Yunita**[1]
[1] School of Computer Science and Technology,    [2] BIT Special Zone
Beijing Institute of Technology
Beijing, China, 100081
{haizhaoyang, liyuan.pan, liuxiabi, liuzhengzheng, mirnayunita}@bit.edu.cn

## Abstract

With the concept of teaching being introduced to the machine learning community, a teacher model start using dynamic loss functions to teach the training of a student model. The dynamic intends to set adaptive loss functions to different phases of student model learning. In existing works, the teacher model 1) merely determines the loss function based on the present states of the student model, *i.e.*, disregards the experience of the teacher; 2) only utilizes the states of the student model, *e.g.*, training iteration number and loss/accuracy from training/validation sets, while ignoring the states of the loss function. In this paper, we first formulate the loss adjustment as a temporal task by designing a teacher model with memory units, and, therefore, enables the student learning to be guided by the experience of the teacher model. Then, with a dynamic loss network, we can additionally use the states of the loss to assist the teacher learning in enhancing the interactions between the teacher and the student model. Extensive experiments demonstrate our approach can enhance student learning and improve the performance of various deep models on real-world tasks, including classification, objective detection, and semantic segmentation scenarios.

## 1  Introduction

In pedagogy study, teachers refine their teaching ability based on student feedback, *e.g.*, exam scores. Students benefited from the enhanced ability of teachers and then achieved high scores on the exam. Both teachers and students are developed in this interaction iteratively and constantly [27–29, 17]. The phenomenon is known as teaching-learning transaction [5], or learning to teach (L2T) in machine learning [7].

In L2T, a teacher model uses a dynamic loss function, which acts as an exam paper, to train and optimize the student model (as shown in Figure 1(a)). However, existing approaches adjust the loss functions by only employing a simple feedforward network as the teacher model, and neglecting the temporal nature of loss function adjustment. The disregarding of the experience accumulation for teachers (*e.g.*, the ability to analyze all previous exam scores for a student), and, therefore, limits the potential of L2T. In addition, in previous works, the teacher model only focuses on the state of the student model, *i.e.*, training iteration number [37], training/validation accuracy [37, 16], training/validation loss[2], and the output of the student model[16, 2, 23]. However, the states of loss functions (*e.g.*, the gradients concerning loss functions) are neglected, which dilutes the benefit of improving the exam paper. In other words, the teacher needs to consider that the question changes of an exam paper also influence the performance of a student.

---

*Equal contribution.  † Corresponding author.

37th Conference on Neural Information Processing Systems (NeurIPS 2023).

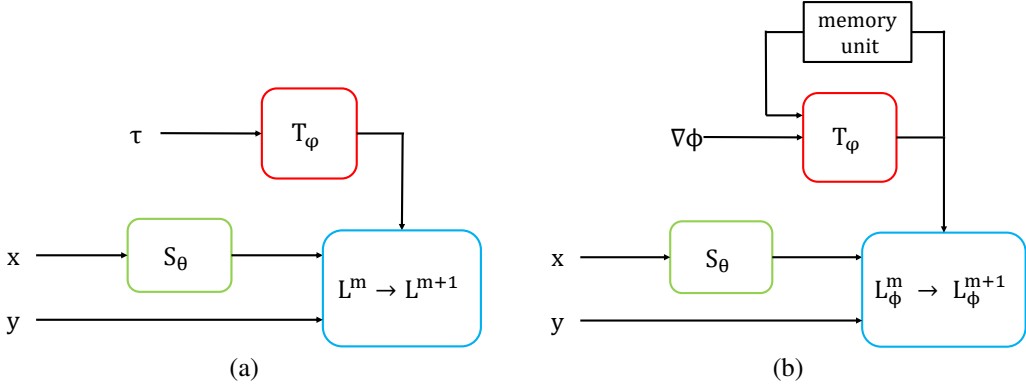

Figure 1: Illustration of the common L2T and our L2T-DLN framework. (a) The common framework in existing L2T works. (b) The framework of our L2T-DLN. Both (a) and (b) contain three models: a student model $S_\theta$ with the parameter $\theta$, a dynamic loss model $L^m$, and a teacher model $T_\varphi$ with the parameter $\varphi$. Here, $m$ denotes the $m^{th}$ iteration, $x$ and $y$ denote the input data of the student and corresponding label, respectively. Different from existing works that only use feedforward networks, we employ a network with a memory unit to enhance the temporal analyzing ability of the teacher. Then, we use the gradient $\nabla\phi$ concerning $L_\phi^m$ by designing a dynamic loss network to provide more information to the teacher model, compared to the state of the student $\tau$.

In this paper, we propose an L2T framework with a Dynamic Loss Network (L2T-DLN), to address the above-mentioned issue (in Figure 1(b)). First, we adopt a Long-Short Term Memory (LSTM) model as the teacher and design a differentiable three-stage asynchronous optimization strategy. Then, to ensure the teacher model can be optimized with the state of loss functions, we design a Dynamic Loss Network (DLN) instead of using the dynamic loss function (DLF). Specifically, we start by optimizing the student model through backpropagation in the first step with a fixed DLN as the loss function. Then, compute the gradient of the validation error of the student model with respect to the DLN. Next, we input this gradient into the teacher model, and the output of the teacher model is used to update the DLN. To achieve the updating of the teacher model, we perform another round of student learning with the updated DLN and obtain the gradient of the validation error of the updated student with respect to the teacher model. Moreover, we analyze how L2T-DLN exploits the negative curvature by using a special alternating gradient descent (AGD) sequence, achieving a differentiable asynchronous optimization.

In summary, the usage of the gradient concerning DLN and the LSTM teacher model both ensure the teacher model captures and maintains short- and long-term temporal information, which can further improve the performance of loss function teaching, compared to feedforward teachers [37, 16, 23].

Our main contributions are 1) design a dynamic loss network-based teaching strategy to let the teacher model learn optimized by the gradient of DLN; 2) use LSTM as the teacher model to update the DLN with the temporal information of the teacher model; and 3) a convergence analysis of the approach, which is treated as a special AGD sequence and has the potential to escape strict saddle points. We conduct extensive experiments on a wide range of loss functions and tasks to demonstrate the effectiveness of our approach.

## 2 Related work

Recent work by L2T [7] provides a comprehensive view of teaching for machine learning, encompassing aspects such as training data teaching, loss function teaching, and hypothesis teaching. In contrast to previous literature on machine teaching [41, 25, 10, 38], L2T breaks the strong assumption regarding the existence of an optimal off-the-shelf student model [37]. Instead, L2T employs automatic techniques to reduce the reliance on prior human knowledge, aligning with principles such as learning to learn and meta-learning [34, 36, 42, 1]. The recent focus of L2T has been mainly on loss function teaching [37, 16, 23, 2] and training data teaching [7, 25, 35, 32, 8].

During the training of a student model, there is a variation in the distribution of predictions where earlier in the training the distribution tends to differ from that at convergence. Consequently, an adaptive loss function is crucial. Existing works [37, 16, 23, 2] formulate the loss adjustment as some independent tasks by performing a multi-layer perceptron (MLP) as the teacher model. The differences lie in the representation of dynamic loss functions and the input information of teacher models. The representation of the dynamic loss function includes the variation of handcrafted loss functions [37, 23] and neural network [16, 2]. The input information contains the training iteration number [37, 16], training/validation accuracy [37, 16], training/validation loss[2], and the output of the student model[16, 2, 23]. In detail, Wu et al. [37] trains a neural network with an attention mechanism to generate a coefficient matrix between the prediction of the student and the ground truth. Huang et al. [16] constructs the teaching-learning framework with reinforcement learning. Their teacher also employs an MLP and generates the policy gradient for a loss network. Liu and Lai [23] utilizes a teacher model to guide the selection and combination of handcrafted loss functions. Baik et al. [2] performs a teacher to generate two weights for each layer of a loss network, and then updates the parameters of the loss network by affine transformation with the two weights.

Assigning weights to different data points has been widely investigated in the literature, where the weights can be either continuous [18] or binary [7]. In detail, Fan et al. [7] proposed a learning paradigm where a teacher model guides the training of the student model. Based on the collected information, the teacher model provides signals to the student model, which can be the weights of training data. Liu et al. [25] leveraged a teaching way to speed up the training, where the teacher model selects the training data balancing the trade-off between the difficulty and usefulness of the data. Fan et al. [8] inputs internal states of the student model, *e.g.*, feature maps of the student model, to the teacher model and obtains the output of the teacher as the weight for corresponding training data.

## 3 Methodology

In this section, we overview our L2T-DLN in Section 3.1, introduce the corresponding framework for student learning in Section 3.2, describe the DLN learning framework in Section 3.3, and discuss teacher learning in Section 3.4.

### 3.1 Overview

Our L2T-DLN is a differentiable teaching framework that enhances the performance of a student model. The L2T-DLN contains stages: (I) student learning, which optimizes a student model $S_\theta$ with parameter $\theta$; (II) DLN learning, which optimizes the DLN $L_\phi$ with parameter $\phi$; (III) teacher learning, which optimizes the teacher model $T_\varphi$ with parameter $\varphi$.

Starting from $S_\theta^0$, in stage (**I**), we optimize the student model $S_\theta^0 \to S_\theta^N$ on training data $x_{train}$ by leveraging the DLN $L_\phi^0$ as the loss function, where $N$ denotes the number of iterations in a student learning stage. In stage (**II**), we compute the error $e_{val}$ of the student model $S_\theta^N$ on validation data $x_{val}$, and then determine the gradient $\nabla\phi^0 = \partial e_{val}/\partial\phi^0$. This gradient is then given to the teacher model, and the DLN is updated as $\phi^1 = \phi^0 + g^0$, where $g^0$ indicates the output of the teacher model $T_\varphi^0$. In stage (**III**), we first train the student model $S_\theta^N \to S_\theta^{2N}$ with the updated DLN $L_\phi^1$. Then, we obtain the validation error $e_{val}$ of the student model $S_\theta^{2N}$ and optimize the teacher model by backpropagation (BP) based on $e_{val}$. The objective function of our L2T-DLN is:

$$(\theta^{(M+K)N}, \phi^M, \varphi^K) \leftarrow (\theta^0, \phi^0, \varphi^0) - \nabla_{(\theta,\phi,\varphi)} \sum_{k=0}^{K} e_{val}(\theta^{2(k+1)N}). \tag{1}$$

Our goal is to achieve $\theta^{(M+K)N}$, $\phi^M$ and $\varphi^K$, where $M$ and $K$ denote the number of iterations of DLN and teacher learning, respectively (details are shown in Figure 2).

### 3.2 Student learning

For a given task, we define the input and output space as $X$ and $Y$, respectively. The student model is then denoted by $S_\theta : X \to Y$. Our student learning involves minimizing the output value of the

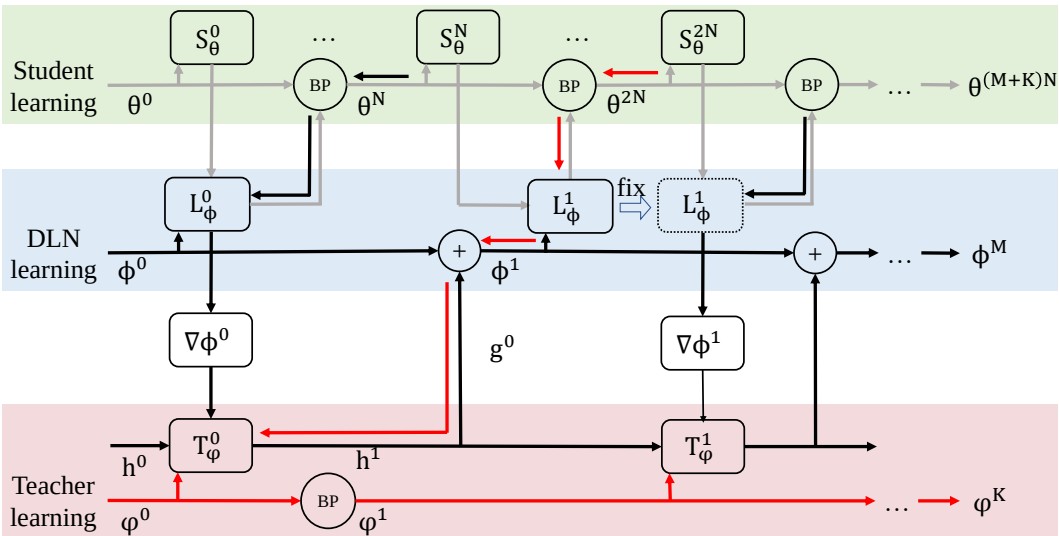

Figure 2: The pipeline of L2T-DLN. Grey, black, and red lines represent the optimization of the student parameter $\theta$, the DLN parameter $\phi$, and the teacher parameter $\varphi$, respectively. We aim at obtaining $\theta^{(M+K)N}$, $\phi^M$, and $\varphi^K$, where $M$ and $K$ denote the number of iterations for DLN and the teacher and $M = K$. There are three stages: student, DLN, and teacher learning. During student learning, a student model is optimized by backpropagation with $L^0_\phi$. During a DLN learning, the teacher $T^0_\varphi$ accept the gradient $\nabla\phi^0$ and output $g^0$ to update $L^0_\phi$. During a teacher learning, we perform another student learning with $L^1_\phi$ and optimize the teacher $\varphi^0$ with the validation error of the updated student.

DLN, i.e., $\underset{\theta \in \Omega}{\arg\min} \sum_{(x,y) \in x_{train}} wL^m_\phi(S_\theta(x), y)$, under a hypothesis space $\Omega$ using training data $x_{train}$. Note, $w$ is a weight parameter of $x$, $L^m_\phi$ is the loss function, and $m$ is the $m^{th}$ iteration for the DLN. During each stage of student learning, we iteratively train the student model $N$ times with $L^m_\phi$. The optimization of the student model during the current stage is:

$$\theta^i = \theta^{i-1} - \eta\partial wL^m_\phi(S^{i-1}_\theta(x), y)/\partial\theta^{i-1}, \; i = \{1, 2, \cdots, N\}, \tag{2}$$

where $\eta$ denotes the learning rate of the student model. In our framework, the $L^m_\phi$ that with learnable parameters are optimized in different student learning stages for providing seemly guidance.

### 3.3 DLN learning

After the student learning stage (e.g., $S^0_\theta \to S^N_\theta$ with $L^0_\phi$), we use the teacher model $T^0_\varphi$ to adjust the DLN parameters $\phi$ (e.g., $\phi^0 \to \phi^1$). To enable the temporal property of DLN, we use an LSTM to transform $\phi$ dynamically:

$$\phi^1 = \phi^0 + \gamma g^0, \; \begin{bmatrix} g^0 \\ h^1 \end{bmatrix} = T^0_\varphi(\nabla\phi^0, h^0), \tag{3}$$

where $\gamma$ denotes the learning rate of DLN and $\nabla\phi^0$ represents the gradient of the validation error $e_{val}$ of $S_\theta$ with respect to $\phi^0$.

Considering the gap between training data and validation data, we employ the Reverse-Mode Differentiation (RMD) to calculate $\nabla\phi^0$. The RMD involves performing the SGD process in reverse order from $N$ to 1, as depicted by the black lines in Figure 2. According to Eq. (2), the gradient of the validation error $e_{val}(S^N_\theta)$ with respect to $\theta^N$ can be calculated as follows:

$$\nabla\theta^N = \partial e_{val}(S^N_\theta)/\partial\theta^N. \tag{4}$$

Then looping backward from $N$. At each step $i = \{N-1, \cdots, 1\}$, the calculation of gradient is shown as:

$$\nabla \theta^{i-1} = \nabla \theta^i - \eta w \frac{\partial^2 L_\phi^0 \left( S_\theta^{i-1}(x), y \right)}{(\partial \theta^{i-1})^2} \nabla \theta^i. \tag{5}$$

At the same time, the gradient of $e_{val}(S_\theta^N)$ with respect to $\phi^0$ is accumulated as:

$$\nabla \phi^0 = \nabla \phi^0 - \eta w \frac{\partial^2 L_\phi^0 \left( S_\theta^{i-1}(x), y \right)}{\partial \theta^{i-1} \partial \phi^0} \nabla \theta^i. \tag{6}$$

Reverting backward from $N$ to $1$, we get $\nabla \phi^0$.

Taking $N = 1$, we rewrite Eq. (6) by using the gradient, e.g., $\nabla \phi^0$, as:

$$\nabla \phi^0 = \eta w \frac{\partial e_{val}}{\partial S_\theta^1(x_{val})} \frac{\partial S_\theta^1(x_{val})}{\partial \theta^1} \left( \frac{\partial \theta^1}{\partial \frac{\partial L_\phi^0(S_\theta^0(x),\, y)}{\partial S_\theta^0(x)}} \frac{\partial \frac{\partial L_\phi^0(S_\theta^0(x_{train}),y)}{\partial S_\theta^0(x)}}{\partial \phi^0} \right) \frac{\partial S_\theta^0(x)}{\partial \theta^0}, \tag{7}$$

where $\nabla \phi^0$ contains the information of both training and validation data, $(\partial S_\theta^1(x_{val})/\partial \theta^1$ and $\partial S_\theta^0(x)/\partial \theta^0)$. As these pieces of information are dependent on each other, they are integrated into temporal changes of $\theta$. The gradient concerning $\phi^0$ achieves holistic information integration throughout the learning process, facilitated by prior knowledge (chain rule). Employing the gradient concerning the loss allows the teacher model to concentrate on capturing and preserving crucial information from gradients, negating the need for supplementary handling of dispersed states. Therefore, compared to the state of the student, *e.g.*, training/validation error, prediction, and numbers of iteration, $\nabla \phi^0$ provides more information to promote deep interaction between DLN, the teacher, and the student.

### 3.4 Teacher learning

To retain the teaching experience, we utilize an LSTM to play the role of teacher. The LSTM utilizes a long-term memory unit, called the cell, to allow itself to maintain effective and long-term dependencies to make predictions, both in the current and future [9]. We employ the coordinate-wise manner [1] to process each value of the input information independently. Different behavior on each coordinate is achieved by using separate activations for each input value. This property leads the student to converge to a good solution.

To evaluate the teacher model, we perform another student learning (*e.g.*, $S_\theta^N \to S_\theta^{2N}$) with the new DLN (*e.g.*, $L_\phi^1$). The RMD is also utilized to calculate the gradient of validation error $e_{val}(S_\theta^{2N})$ with respect to the teacher model parameters $\varphi^0$. We represent the computation using red lines in Figure 2. To obtain $\nabla \varphi^0$, we loop the SGD process backward from $2N$ to $N+1$ with the updated DLN $L_\phi^1$ according to Eq. (4), (5) and (6). At each step $i = \{2N-1, \cdots, N+1\}$, the gradient $\nabla \varphi^0$ is updated as:

$$\nabla \varphi^0 = \nabla \varphi^0 - \eta \gamma w \frac{\partial^3 L_\phi^1 \left( S_\theta^{i-1}(x_{train}), y \right)}{\partial \theta^{i-1} \partial \phi^1 \partial \varphi^0} \nabla \theta^i. \tag{8}$$

The process of L2T-DLN is summarized in Algorithm 1.

## 4  Convergence Analysis

Since our L2T-DLN is updated asynchronously, we can only access partial second-order information at each training stage. For example, given a quadratic objective function, while fixing one part of L2T-DLN, the problem is strongly convex with respect to the other part, but the entire problem is nonconvex. Even if the iterates converge for each part to the corresponding minimum points, the stationary point could still be a saddle point for the overall objective function [26]. Therefore, the analysis of how L2T-DLN exploits the negative curvature is necessary.

The Alternating Gradient Descent (AGD) [4] algorithm only updates partial variables of vector, *e.g.*, $v = \{\theta, \varphi + \phi\}$, which belongs to a subset of the feasible set. From the mean value theorem, we can

---

**Algorithm 1** Obtaining the optimal student, DLN and teacher in L2T-DLN.

---

1: **Hyperparameter:** length of the student training $N$, total number of iterations for DLN learning $M$, total number of iterations for teacher learning $K$.
2: **Input:** Random initialization parameter $\theta^0$, $\phi^0$, and $\varphi^0$.
3: $m = 0$
4: **for** $k = (1, \cdots, K)$ **do**
5:     **for** $i = ((m+k)N+1, \cdots, (m+k+1)N)$ **do**       ▷ A student learning stage
6:         Conduct student model training step via Eq. (2) ($S_\theta^{i-1} \to S_\theta^i$).
7:     **end for**
8:     $\nabla\phi^m = 0$, compute $\nabla\theta^{(m+k+1)N}$ via Eq. (4).
9:     **for** $i = ((m+k+1)N, \cdots, (m+k)N+1)$ **do**  ▷ Reversely calculating the gradient $\nabla\phi$
10:         Update $\nabla\phi^m$ via Eq. (6).
11:         Compute $\nabla\theta^i$ via Eq. (5).
12:     **end for**
13:     update $\phi^m \to \phi^{m+1}$ via Eq. (3).
14:     $m = m + 1$                                 ▷ Updating $m$
15:     **for** $i = ((m+k)N+1, \cdots, (m+k+1)N)$ **do**       ▷ A student learning stage
16:         Conduct student model training step via Eq. (2) ($S_\theta^{i-1} \to S_\theta^i$).
17:     **end for**
18:     $\nabla\varphi^k = 0, \nabla\phi^m = 0$, compute $\nabla\theta^{(m+k+1)N}$ via Eq. (4)
19:     **for** $i = ((m+k+1)N, \cdots, (m+k)N+1)$ **do**  ▷ Reversely calculating the gradient $\nabla\varphi$
20:         update $\nabla\phi^m$ via Eq. (6).
21:         Update $\nabla\varphi^k$ via Eq. (8).
22:         Compute $\nabla\theta^i$ via Eq. (5).
23:     **end for**
24:     Update $\varphi^k \to \varphi^{k+1}$ using $\nabla\varphi^k$ via gradient based optimization algorithm.
25: **end for**
26: **output:** $\theta^{(M+K)N}, \phi^M, \varphi^K$

---

express the AGD rule of updating variables by assuming $v^{(0)} = 0$ as follows, with $e$ denoting the validation error.

$$
\begin{aligned}
v^{k+1} &= v^k - \eta \begin{pmatrix} \nabla_1 e(v_1^{2kN}, v_2^k) \\ \nabla_2 e(v_1^{2(k+1)N}, v_2^k) \end{pmatrix} \\
&= v^k - \eta \int_0^1 \mathcal{H}_l^k dv^{2(k+1)N} - \eta \int_0^1 \mathcal{H}_u^k dv^k,
\end{aligned}
\tag{9}
$$

where $\mathcal{H}_l^k \triangleq \begin{bmatrix} 0 & 0 \\ \nabla_{21}^2 e(v_1^{2(k+1)N}, v_2^k) & 0 \end{bmatrix}$ and $\mathcal{H}_u^k \triangleq \begin{bmatrix} \nabla_{11}^2 e(v_1^{2kN}, v_2^k) & \nabla_{12}^2 e(v_1^{2kN}, v_2^k) \\ 0 & \nabla_{22}^2 e(\theta v_1^{2(k+1)N}, \theta v_2^k) \end{bmatrix}$.

The right-hand side of Eq. (9) not only contains the second order information of the previous point, i.e., $[v_1^{2kN}, v_2^k]$, but also the one of the most recently updated point, i.e., $[v_1^{2(k+1)N}, v_2^k]$.

Different from traditional AGD, the dynamic system in L2T-DLN takes the first-order information to update the student and the second-order information to update the teacher. Specifically, $\nabla_1 e(v_1^{2kN}, v_2^k) = \nabla e(v_1^{2kN}, v_2^k)$ and $\nabla_2 e(v_1^{2(k+1)N}, v_2^k) = \nabla^2 e(v_1^{2(k+1)N}, v_2^k)$. These represent the main challenges in understanding the behavior of the sequence generated by the AGD algorithm.

Although the higher-order information is divided into two parts, we can still characterize the recursion of the iterates around strict saddle points $v^*$. We can also split $\mathcal{H}$ as two parts, which are

$$
\mathcal{H}_u = \begin{bmatrix} \nabla_{11}^2 e(v^*) & \nabla_{12}^2 e(v^*) \\ 0 & \nabla_{22}^2 e(v^*) \end{bmatrix}, \quad \mathcal{H}_l = \begin{bmatrix} 0 & 0 \\ \nabla_{21}^2 e(v^*) & 0 \end{bmatrix},
\tag{10}
$$

and obviously, we have $\mathcal{H} = \mathcal{H}_u + \mathcal{H}_l$.

Then recursion Eq. (9) can be written as

$$
v^{2(k+1)N} + \eta \mathcal{H}_l v^{2(k+1)N} = x^k - \eta \mathcal{H}_u v^k - \eta \triangle_u^k v^k - \eta \triangle_l^k v^{2(k+1)N},
\tag{11}
$$

where $\triangle_u^k \triangleq \int_0^1 (\mathcal{H}_u^k(v) - \mathcal{H}_u)dv$, $\triangle_l^k \triangleq \int_0^1 (\mathcal{H}_l^k(v) - \mathcal{H}_l)dv$. However, it is still unclear from Eq. (11) how the iteration evolves around the strict saddle point. To highlight ideas, let us define

$$M \triangleq I + \eta\mathcal{H}_l, \ G \triangleq I - \eta\mathcal{H}_u. \tag{12}$$

It can be observed that $M$ is a lower triangular matrix where the diagonal entries are all 1s; therefore it is invertible. After taking the inverse of matrix $M$ on both sides of Eq. (11), we can obtain

$$v^{k+1} = M^{-1}Gv^k - \eta M^{-1} \triangle_u^k v^k - \eta M^{-1} \triangle_l^k v^{2(k+1)N}. \tag{13}$$

Our goal of analyzing the recursion of $v^k$ becomes to find the maximum eigenvalue of $M^{-1}G$. With the help of the matrix perturbation theory, we can quantify the difference between the eigenvalues of matrix $\mathcal{H}$ that contains the negative curvature and matrix $M^{-1}G$ that we are interested in analyzing. With the gradient Lipschitz constants $\{\tilde{C}_k\}$, we set $L_{max} \triangleq max\{C_k, \tilde{C}_k, \forall k\} \leq C$ and give the following conclusion.

**Conclusion 1.** Let $\mathcal{H} \triangleq \triangledown^2 e(x)$ denote the Hessian matrix at an $\epsilon-$second-order stationary solution (SS2) $v^*$ where $\lambda_{min}(\mathcal{H}) \leq -\gamma$ and $\gamma > 0$. We have

$$\lambda_{max}(M^{-1}G) > 1 + \frac{\eta\gamma}{1 + C/C_{max}}. \tag{14}$$

The proof of Conclusion 1 contains the following steps:

**Step 1.** (Lemma 1 [26]) Giving a generic sequence $u$ generated by AGD ($v^k \in u$). As long as the initial point of $u^k$ is close to saddle point $\tilde{v}^k$, the distance between $u^k$ and $\tilde{v}^k$ can be upper bounded by using the $\rho-$Hessian Lipschitz continuity property.

**Step 2.** Leveraging the negative curvature around the strict saddle point, we can project the $u^k$ onto the two subspaces, where the first subspace is spanned by the eigenvector of $M^{-1}G$ and the other one is spanned by the remaining eigenvectors. We use two steps to show $\lambda_{max}(M^{-1}G) > 1$: 1) we show that all eigenvalues of $Q(\lambda) = [G - \lambda M]$ are real; 2) $\exists \lambda > 1, det(Q(\lambda)) = 0$.

Conclusion 1 illustrates that there exists a subspace spanned by the eigenvector of $M^{-1}G$ whose eigenvalue is greater than 1, indicating that the sequence generated by AGD can still potentially escape from the strict saddle point by leveraging such negative curvature information (more can be found in supplementary materials).

## 5 Experiments

### 5.1 Experimental setup

**Datasets.** We evaluate our method on three tasks, *i.e.*, image classification, objective detection, and semantic segmentation. For the image classification, we use three datasets: CIFAR-10 [20], CIFAR-100 [21], and ImageNet [33]. CIFAR-10 and CIFAR-100 contain 50000 training and 10000 testing images with 10-class and 100-class separately. ImageNet is a 1000-class task that contains 1281167 training and 50000 testing pairs. For the objective detection, we use MS-COCO dataset [22], which contains 82783, 40504, and 81434 pairs in the training, validation, and testing set separately. For the semantic segmentation, we choose PASCAL VOC 2012 [6]. Following the procedure of Zhao et al. [40], we use augmented data with the annotation of Hariharan et al. [11], resulting in 10582, 1449, and 1456 images for training, validation, and testing.

**Evaluation metrics.** In the classification, we use the accuracy on the testing set of each dataset [37, 23]. In the objective detection, we use the mean of Average Precision (mAP) [31] to evaluate the student model on the testing set of MS-COCO [22]. In the semantic segmentation, we use Mean Intersection over Union (mIoU) [40] to evaluate the student model on the testing set of VOC [6, 11].

**Baseline methods.** For the classification, we employ several popular loss functions, including fixed loss functions such as Cross Entropy loss (CE), the large-margin softmax loss (L-M softmax) [24], and the smooth 0-1 loss function (Smooth) [30] as well as dynamic loss functions, namely the adaptive robust loss function (ARLF) [3], the L2T-DLF loss function [37], stochastic loss function (SLF) [23] and ALA [16]. For objective detection, we compare our approach with the objective function set by YOLO-v3 [31]. For the semantic segmentation, we compare our approach with the objective function set by PSPNet [40].

Table 1: Results on datasets CIFAR-10 [20], CIFAR-100 [21] and ImageNet [33] for the classification task. All experiments are implemented with the same settings. The best results are highlighted in bold.

| Method | CIFAR-10 [20] | | | | CIFAR-100 [21] | | | ImageNet [33] | length |
| | ResNet8 | ResNet20 | ResNet32 | WRN | ResNet8 | ResNet20 | ResNet32 | NASNet-A | |
|---|---|---|---|---|---|---|---|---|---|
| CE | 87.6 | 91.3 | 92.5 | 96.2 | 60.2 | 67.7 | 69.6 | 73.5 | - |
| Smooth [30] | 87.9 | 91.5 | 92.6 | 96.2 | 60.5 | 68.0 | 69.9 | - | - |
| L-M Softmax [24] | 88.7 | 92.0 | 93.0 | 96.3 | 61.1 | 68.4 | 70.4 | - | - |
| L2T-DLF [37] | 89.2 | 92.4 | 93.1 | 96.6 | 61.7 | 69.0 | 70.8 | - | 1 |
| ARLF [3] | 89.5 | 91.5 | 92.2 | 95.9 | 60.2 | 67.8 | 69.9 | - | - |
| SLF [23] | 89.8 | 93.0 | 93.6 | **97.1** | 62.7 | 69.9 | 71.5 | - | - |
| ALA [16] | - | - | 93.2 | 96.7 | 62.2 | 69.5 | 70.9 | **74.6** | 200 [15] |
| Ours | **90.7 ± 0.06** | **93.4 ± 0.18** | **93.8 ± 0.20** | 96.7 ± 0.09 | **63.5 ± 0.07** | **70.4 ± 0.03** | **72.0 ± 0.11** | 74.2 | 25 |

**Implementation details.** In all experiments, we optimize student models using standard stochastic gradient descent (SGD) with a learning rate of 0.1. The teacher model is trained with Adam, utilizing a learning rate of 0.001. The learning rate of DLN is set to 0.001. The teacher model is trained for 10 epochs, with redividing the training and validation data after each epoch. The validation errors in each task are explicitly reported. Our teacher model comprises a four-layer LSTM [14] with 64 neurons in the first three layers and 1 neuron in the final layer. We utilize a 1-vs-1 approach (details in supplementary materials) to process the student model's output in both classification and segmentation. We present DLN architecture for each task and ensure reliable evaluation by conducting 5 random restarts, using average results for comprehensive comparisons.

## 5.2 Results

**Image classification.** For CIFAR-10 and CIFAR 100, we follow the SLF [23] and use architectures that include ResNet [12], and Wide-ResNet(WRN) [39] as the student model. For ImageNet, we follow the ALA [16] and use the identical NASNet-A [43]. In each experiment, the batch sizes for training and validation are set to 25 and 100, respectively. We perform a five-layer fully connected network, which contains 40 neurons in each hidden layer and 1 neuron in the output layer, as the DLN. The activation function for each hidden layer is set to Leaky-ReLU. The validation error is computed by CE.

Table 1 reports the performance of each loss function. Our approach achieves the best results on CIFAR-10 with ResNet8, ResNet20, and ResNet32, achieving $90.70\%, 93.40\%$, and $93.81\%$, respectively. On CIFAR-100, our method also outperforms baselines with an overall accuracy of $63.50\%, 70.47\%$, and $72.06\%$ for ResNet8, ResNet20, and ResNet32, respectively. For WRN, our approach achieves the second-best performance, following the SLF method. The results on ImageNet illustrate that L2T-DLN improves the accuracy of the baseline by $0.7\%$. On ImageNet, our DLN demonstrates the second-best performance. The performance of ALA benefits from the larger length of a student learning stage ALA set (200) compared with ours (25). The ablation showed that the size of the length is positively correlated with the test accuracy and computational consumption (see Table 5).

**Objective detection.** In the task of objective detection, the YOLO-v3 model with a backbone of darknet-53 [31] is used in this experiment. The traditional loss in the YOLO model is a multi-part loss function, *i.e.*, $\lambda_{cls}\ell_{cls} + \lambda_{conf}\ell_{conf} + \lambda_{loc}\ell_{loc}$. $\ell_{cls}, \ell_{conf}$ and $\ell_{loc}$ are detailed in supplementary materials. Redmon and Farhadi [31] set $\lambda_{cls} = \lambda_{conf} = \lambda_{loc} = 1$. In our experiment, our L2T-DLN learns to set these weights dynamically with a single-layer perceptron as DLN. The backbone of the YOLO is pre-trained on ImageNet, and we finetune the header of the YOLO. Specifically, the objective function of the student model is set to $DLN([\ell_{cls}, \ell_{conf}, \ell_{loc}])$. The validation error is computed by $\ell_{cls} + \ell_{conf} + \ell_{loc}$. The batch sizes for training and validation are set to 2 and 8, respectively. The length of student learning is set to 2. We take the training set and 35000 images of the validation set to train our L2T-DLN with an input size of 416*416. From Table 2, our L2T-DLN has more than $1.6\%$ improvement with the baseline on mAP.

**Semantic segmentation.** The objective function of PSPNet [40] is set to $CE(p, y) + 0.4 * CE(aux_p, y)$, where $p, aux_p$, and $y$ denote the output of the master branch, the auxiliary branch of PSPNet, and the ground truth, respectively. For PSPNet with L2T-DLN, the objective function is set to $DLN(p, y) + 0.4 * DLN(aux_p, y)$, where the architecture of DLN is the one used in classification tasks. The validation error is computed by $CE(p, y) + 0.4 * CE(aux_p, y)$. The batch sizes for

Table 2: Objective detection on COCO [22]. DLN and original losses in YOLOV3.

| Detectors | Size | mAP | FPS |
|---|---|---|---|
| YOLOV3 [31] | 416 | 55.3 | 35 |
| YOLOV3-ours | 416 | 56.9 | 35 |

Table 3: Segmentation on VOC [6, 11]. DLN and original losses in PSPNet.

| Method | mIoU |
|---|---|
| PSPNet [40] | 82.6 |
| PSPNet-ours | 82.9 |

training and validation are set to 2 and 8, respectively. The length of student learning is set to 2. Table 3 shows that our L2T-DLN improves $0.3\%$ compared with the baseline on mIoU.

## 5.3 Ablations

In this subsection, we conduct ablation studies on CIFAR-10 [20] using ResNet8 to analyze the L2T-DLN synthetically. We specifically examine the proportion of training and validation data, the length of student learning stage ($N$), the wrong learning rate setting, and the influence of the LSTM teacher. Furthermore, we provide the visualization of DLN at different learning stages in MNIST and CIFAR10 tasks. We assess the impact of each component by computing the test accuracy of the student model after optimizing the teacher model for 10 epochs.

**The proportion of training and validation data.** In L2T-DLN, the training dataset is divided into two sets: validation data and training data, with validation data serving as an unbiased estimator for model generalization. After each epoch, the dataset is redivided, allowing samples used in the validation data to be included in the training data, and vice versa. The **validation ratio** represents the fraction of training dataset samples exclusively used for validation. This study explores different training-validation data separations. Table 4 results indicate our performance remains stable across varying ratios due to the teacher's ability to capture short- and long-term dependencies. To make a trade-off between computational cost and accuracy, we set the ratio= $50\%$ for all our experiments.

Table 4: Results on different validation ratios ranging from $10\%$ to $90\%$ to show the impact of ratios.

| Ratio | 10% | 25% | 50% | 75% | 90% |
|---|---|---|---|---|---|
| Accuracy | 90.41 | 90.53 | 90.70 | 90.68 | 90.35 |

**The length of student learning.** The computation of higher-order gradients in L2T-DLN (Eq. (6) and (8)) is computationally intensive and should be highlighted. Thus, this study explores the influence of the length of student learning ($N$) on the test accuracy and computational load in CIFAR-10 experiments using ResNet8. As shown in Table 5, the findings reveal that the test accuracy increases with the length of student learning. To make a trade-off between performance and computational cost, we suggest that a maximum length of 25 should be set for student learning. Overall, the study concludes that L2T-DLN has the potential to further improves the performance of student model with sufficient computing resources.

Table 5: Results on different lengths ranging from 1 to 75 to show the impact of the length. Time denotes the time consumption of a round of teacher learning.

| Length | 1 | 5 | 10 | 15 | 25 | 50 | 75 |
|---|---|---|---|---|---|---|---|
| Accuracy | 81.40 | 87.07 | 89.95 | 90.18 | 90.70 | 90.73 | 90.74 |
| Time | 1s | 3s | 5.8s | 7.9s | 13.4s | 32s | 76.7s |

**The influence of an LSTM teacher.** As introduced above, the teacher model is similar to an optimization algorithm. Then we perform various optimizers, including Adam [19], SGD, RMSProp [13], and the LSTM teacher, to optimize the DLN and present the results in Table 6. Compared with ADAM, SGD, and RMSProp, our teacher can improve the performance of the student by $0.48\%$, $1.6\%$, and $0.53\%$. We can conclude that 1) algorithms that can use the historical information, *e.g.*, momentum, perform well; 2) the adaptability to capture and maintain short- and long-term dependencies can further enhance the loss function teaching, compared to handcrafted methods, *e.g.*, exponentially weighted moving average [13] and moment estimation [19].

Table 6: Results on different optimizers to show the effectiveness of the LSTM teacher.

| Optimizer | Adam [19] | SGD | RMSProp [13] | LSTM |
|-----------|-----------|-------|--------------|-------|
| Accuracy | 90.17 | 89.05 | 90.12 | 90.65 |

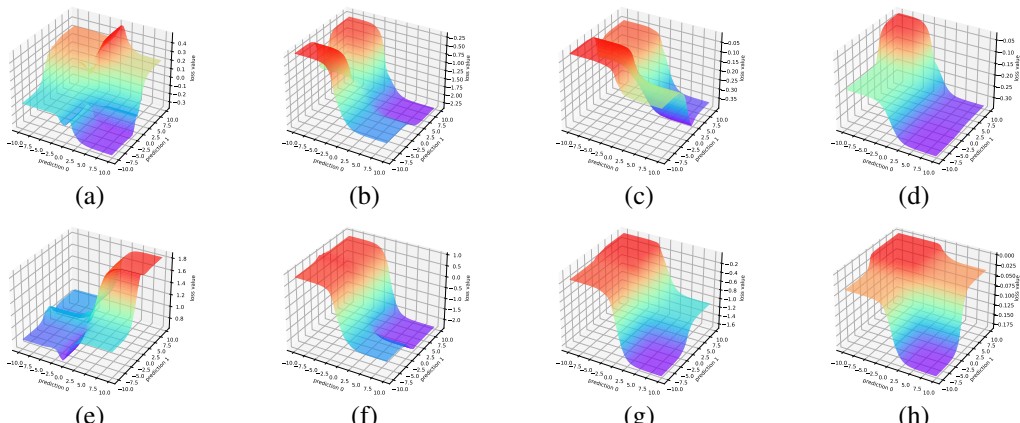

Figure 3: Visualization of DLN loss value at different training stages in the MNIST-LeNet task (a-d) and CIFAR10-ResNet8 task (e-h). (a) & (e) initialized DLN, (b) & (f) DLN finished second teacher learning epoch, (c) & (g) DLN finished fifth teacher learning epoch, (d) & (h) DLN finished tenth teacher learning epoch (final). The X-, Y-, and Z-axis are the prediction for 0-category (denoted as prediction0), the prediction for 1-category (denoted as prediction1), and the loss value of DLN, respectively. We set the 0-category as the correct category and the 1-category as the wrong category. We can observe that the output value range of DLN initially expands and subsequently contracts. Specifically, the range shifts from (-0.3, 0.4) to (-2.25, -0.25) to (-0.3, -0.05) in MNIST, and from (0.8, 1.8) to (-2, 1) to (-0.175, 0) in CIFAR10.

**Visualization.** We visualize the loss value of DLN on MNIST and CIFAR-10 separately in Figure 3, which illustrates the capacity of L2T-DLN to adapt to the evolving states of students to attain improved performance. The DLN is initialized with the Kaiming normal initialization with LeakyReLU activations.

## 6  Conclusions

This paper introduces L2T-DLN, an adaptive model for various stages of student learning. Technically, We propose a differentiable three-stage teaching framework, asynchronously optimizing the student, DLN, and teacher. An LSTM teacher dynamically captures and retains experiences during DLN learning. Additionally, we assess L2T-DLN's ability to navigate strict saddle points using the negative curvature of their Hessian matrix. Experiments demonstrate our DLN outperforming specially designed loss functions. Nevertheless, our approach demands significant computational resources for high-order derivatives, which we aim to mitigate in future work.

## 7  Acknowledgment

This work was supported in part by National Natural Science Foundation of China (62302045, 82171965), Clinical and Translational Medical Research Fund of the Chinese Academy of Medical Sciences (2020-I2M-C&T-B-072), and Beijing Institute of Technology Research Fund Program for Young Scholars.

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
