# Supplementary Materials for the Paper "L2T-DLN: Learning to Teach with Dynamic Loss Network"

**Zhaoyang Hai** [1,*], **Liyuan Pan** [1,2,*,†], **Xiabi Liu** [1,*,†], **Zhengzheng Liu**[1], **Mirna Yunita**[1]

[1] School of Computer Science and Technology, [2] BIT Special Zone
Beijing Institute Of Technology
Beijing, China, 100081
{haizhaoyang, liyuan.pan, liuxiabi, liuzhengzheng, mirnayunita}@bit.edu.cn

In this supplementary material, we provide the proofs of convergence analysis in Section 1, 1-vs-1 transformation employed in the classification and semantic segmentation tasks in Section 2, the coordinate-wise and the preprocessing method of the LSTM teacher in Section 3, the loss functions of YOLO-v3 in Section 4, more experiments of image classification in Section 5, and the inferences of semantic segmentation in Section 6.

## 1 Convergence analysis

### 1.1 Preliminaries

**Definition 1.** A differentiable function $e(\cdot)$ is L-smooth with gradient Lipschitz constant $C$ (uniformly Lipschitz continuous), if $\| \bigtriangledown e(x) - \bigtriangledown e(y)\| \leq C\|x - y\|, \forall x, y$. The function is called block-wise smooth with gradient Lipschitz $C_i$, if

$$\| \bigtriangledown_i e(x_{-i}, x_i) - \bigtriangledown_i e(x_{-i}, x_i')\| \leq C_i\|x_i - x_i'\|, \forall x, x' \tag{1}$$

or with gradient Lipschitz constants $\{\tilde{C}_i\}$, if

$$\| \bigtriangledown_i e(x_{-i}, x_i) - \bigtriangledown_i e(x_{-i}', x_i)\| \leq \tilde{C}_i\|x_{-i} - x_{-i}'\|, \forall x, x' \tag{2}$$

Further, Let $G_{max} \triangleq max\{C_i, \tilde{C}_i, \forall k\} \leq C$.

**Definition 2.** For a differentiable function $e(\cdot)$, if $\| \bigtriangledown e(x)\| = 0$, then $x$ is a first-order stationary solution (SS1). If $\| \bigtriangledown e(x)\| \leq \epsilon$, then $x$ is an $\epsilon$-first-order stationary point.

**Definition 3.** For a differentiable function $e(\cdot)$, if $x$ is a SS1, and there exists $\epsilon > 0$ so that for any $y$ in the $\epsilon$-neighborhood of $x$, we have $e(x) \leq e(y)$, then $x$ is a local minimum. A saddle point $x$ is an SS1 that is not a local minimum. If $\lambda_{min}(\bigtriangledown^2 e(x)) < 0$, x is a strict (non-degenerate) saddle point.

**Definition 4.** A twice-differentiable function $e(\cdot)$ is $\rho$-Hessian Lipschitz if

$$\| \bigtriangledown^2 e(x) - \bigtriangledown^2 e(y)\| \leq \rho\|x - y\|, \forall x, y \tag{3}$$

**Definition 5.** For a $\rho$-Hessian Lipschitz function $e(\cdot)$, $x$ is a second-order stationary solution (SS2) if $\| \bigtriangledown e(x)\| = 0$ and $\lambda_{min}(\bigtriangledown^2 e(x)) \geq 0$. If the following holds

$$\| \bigtriangledown e(x)\| \leq \epsilon, \ and \ \lambda_{min}(\bigtriangledown^2 e(x)) \geq -\gamma \tag{4}$$

where $\epsilon, \gamma > 0$, then $x$ is a $(\epsilon, \gamma)$ SS2.

**Condition 1.** An $\epsilon-$second order stationary point $\tilde{v}^k$ satisfies the following conditions:

$$\sum_{i=1}^{2} \| \bigtriangledown_i e(\tilde{h}_{-i}^k, \tilde{v}_i^k)\|^2 \leq g_{th}^2 \ and \ \lambda_{min}(\bigtriangledown^2 e(\tilde{v}^k)) \leq -\gamma \tag{5}$$

---

[*]Equal contribution. [†] Corresponding author.

37th Conference on Neural Information Processing Systems (NeurIPS 2023).

where $g_{th} \triangleq \frac{\mathcal{G}}{2\mathcal{K}}$. $\mathcal{G}$ denotes the size of the gradient, $\mathcal{K} \triangleq \frac{C_{max}}{\gamma} \geq 1$.

Based on the $\rho$-Hessian Lipschitz property, we can quantify $\| \triangle^k \|$ that is upper bounded by the difference of iterates. By exploiting the negative curvature of the Hessian matrix at saddle point $v^*$, we can project the iterate onto the direction $\vec{d}$ where the eigenvalue of $I - \eta\mathcal{H}$ is greater than 1. This leads to the fact that the norm of the iterates projected along direction $\vec{d}$ will be increasing exponentially as the algorithm proceeds around point $v^*$, implying the sequence generated by Gradient Descent is escaping from the saddle point. The details of characterizing the convergence rate have been analyzed previously in [9].

## 1.2 Main proofs

**Conclusion 1.** Let $\mathcal{H} \triangleq \bigtriangledown^2 e(v^*)$ denote the Hessian matrix at an $\epsilon-$second-order stationary solution (SS2) $v^*$ where $\lambda_{min}(\mathcal{H}) \leq -\gamma$ and $\gamma > 0$. We have

$$\lambda_{max}(M^{-1}G) > 1 + \frac{\eta\gamma}{1 + C/C_{max}}. \tag{6}$$

The proof of Conclusion 1 contains the following steps:

**Step 1.** (Lemma 1 [14]) Giving a generic sequence $u$ generated by Alternating Gradient Descent ($v^k \in u$). As long as the initial point of $u^k$ is close to saddle point $\tilde{v}^k$, the distance between $u^k$ and $\tilde{v}^k$ can be upper bounded by using the $\rho-$Hessian Lipschitz continuity property.

**Step 2.** Leveraging the negative curvature around the strict saddle point, we can project the $u^k$ onto the two subspaces, where the first subspace is spanned by the eigenvector of $M^{-1}G$ and the other one is spanned by the remaining eigenvectors. We use two steps to show $\lambda_{max}(M^{-1}G) > 1$: 1) we show that all eigenvalues of $Q(\lambda) = [G - \lambda M]$ are real; 2) $\exists \lambda > 1, det(Q(\lambda)) = 0$.

For **step 1**, we provide the lemma 1 and the corresponding proof.

**Lemma 1** Consider $\tilde{v}^k$ that satisfies Condition 1 and a generic sequence $u^k$ generated by AGD. For any constant $\hat{c} \geq 2, \delta \in (0, \frac{d\mathcal{K}}{e}]$, when initial point $u^0$ satisfies

$$\|u^0 - \tilde{v}^k\| \leq 2r, \ r \triangleq \frac{\eta C_{max}S}{\mathcal{K}log(\frac{d\mathcal{K}}{\delta})\mathcal{P}_1} \tag{7}$$

where $\mathcal{P}_1 = (1 + \frac{C}{C_{max}})$. With $G \triangleq \min_{k}\{inf\{k|\hat{e}_{u^0}(u^k) - e(u^0) \leq -3\mathcal{F}\}, \hat{c}\mathcal{T}\}$, where $\mathcal{F}$ accounts for the objective value, and $\mathcal{T}$ for the number of iterations, there exists constants $c^1_{max}, \hat{c}$ such that for any $\eta \leq c^1_{max}/C_{max}$, the iterates generated by L2T-DLN satisfy $\|u^k - \tilde{v}^k\| \leq 5\hat{c}\mathcal{S}, \forall t < G$.

**Proof.** Without loss of generality, let $u^0$ be the origin, *i.e.*, $u^0 = 0$. According to the AGD update rules, we have

$$u^{k+1} = u^k - \eta \begin{bmatrix} \bigtriangledown_1 e(u_1^k, u_2^k) \\ \bigtriangledown_2 e(u_1^{k+1}, u_2^k) \end{bmatrix}. \tag{8}$$

Then, we introduce the AGD update rules, we have

$$\|u^k\|, \ \forall k < K. \tag{9}$$

When $k = 0$, we have $u^0 = 0$, Eq. (9) is true. Suppose Eq. (9) is true for the case where $t \leq k$. We will show that Eq. (9) is also true for the case where $t = k + 1$.

First, we show the upper bound of $\|u^{k+1} - u^k\|$. According to the Taylor expansion and $\rho$-Hessian Lipschitz continuity, we have

$$f(u^k) \leq f(u^0) + \bigtriangledown f(u^0)^T (u^k - u^0) + \frac{1}{2}(u^0 - u^k)^T \bigtriangledown^2 f(u^0)(u^0 - u^t) + \frac{\rho}{6}\|u^k - u^0\|^3. \tag{10}$$

Comparing with the definition of $\hat{f}_{u^0}(u^k)$, we have

$$|f(u^k) - \hat{f}_{u^0}(u^k)| \leq \frac{1}{2}(u^0 - u^k)^T (\bigtriangledown^2 f(u^0) - \mathcal{H})(u^0 - u^k) + \frac{\rho}{6}\|u^k - u^0\|^3$$

$$\leq \frac{\rho}{2}\|u^0 - \tilde{v}^k\|\|u^k - u^0\|^2 + \frac{\rho}{6}\|u^k - u^0\|^3.$$

According to the definition of $G$, we have $f(u^0) - \hat{f}_{u^0}(u^k) \leq 3\mathcal{F}$ for all $k < K$, which implies that

$$f(u^0) - f(u^k) \leq |f(u^0) - \hat{f}_{u^0}(u^k)| + |\hat{f}_{u^0}(u^k) - f(u^k)|$$

$$\leq 3\mathcal{F} + \frac{\rho}{2}\|\tilde{v}^k - u^0\|\|u^k - u^0\|^2 + \frac{\rho}{6}\|u^k - u^0\|^3$$

$$\leq 3\mathcal{F} + \frac{\rho}{2}\frac{\eta L_{max}\mathcal{S}}{\mathcal{K}log(\frac{d\mathcal{K}}{\delta})\mathcal{P}_1}(5\hat{c}\mathcal{S})^2 + \frac{\rho}{6}(5\hat{c}\mathcal{S})^3$$

$$\leq 3\mathcal{F} + ((5\hat{c})^2/4 + (5\hat{c})^3/6)\rho\mathcal{S}^3$$

$$\leq 3\mathcal{F} + \eta L_{max}(5\hat{c})^3\mathcal{F}\mathcal{P}_2^{-1} \leq 4\mathcal{F} \qquad (11)$$

where $d$ denotes the dimension of parameters. We use $c_{max} = \mathcal{P}_2/(5\hat{c})^3$ and $\eta \leq c_{max}/L_{max}$. Given

$$f(u^{k+1}) \leq f(u^k) - \frac{\eta}{2}(\|\nabla_1 f(u_1^k, u_2^k))\|^2 + \|\nabla_2 f(u_1^{k+1}, u_2^k))\|^2), \forall k < K. \qquad (12)$$

For simplification of experession, we define $z_{-1}^k \triangleq u_2^k$ and $z_{-2}^k \triangleq u_1^{k+1}$, where $k < K$. For $t = 0, \cdots, k$, we have

$$f(u^k) \leq f(u^0) - \sum_{t=0}^{k-1}\sum_{i=1}^{2}\frac{\eta}{2}\|\nabla_i e(z_{-i}^t, u_i^t)\|^2. \qquad (13)$$

According Eq. (11) and Eq. (13), we know that

$$\sum_{t=0}^{k-1}\sum_{i=1}^{2}\frac{\eta}{2}\|\nabla_i e(z_{-i}^t, u_i^t)\|^2 \leq 4\mathcal{F}, \qquad (14)$$

or

$$\max_{t}\sum_{i=1}^{2}\|\nabla_i f(z_{-i}^t, u_i^t)\|^2 \leq 4\mathcal{F}, \ t \leq k-1. \qquad (15)$$

According to Eq. (8), we have

$$\|u^{k+1} - u^k\|^2$$

$$= \eta^2\sum_{i=1}^{2}\|\nabla_i e(z_{-i}^k, u_i^k)\|^2$$

$$= 2\eta^2\sum_{i=1}^{2}\|\nabla_i e(z_{-i}^k, u_i^k) - \nabla_i e(z_{-i}^{k-1}, u_i^{k-1})\|^2 + 2\eta^2\sum_{i=1}^{2}\|\nabla_i e(z_{-i}^{k-1}, u_i^{k-1})\|^2$$

$$= 2\eta^2(2\sum_{i=1}^{2}\|\nabla_i e(z_{-i}^k, u_i^k) - \nabla_i e(z_{-i}^{k-1}, u_i^k)\|^2 + 2\sum_{i=1}^{2}\|\nabla_i e(z_{-i}^{k-1}, u_i^k) - \nabla_i e(z_{-i}^{k-1}, u_i^{k-1})\|^2)$$

$$+ 2\eta^2\sum_{i=1}^{2}\|\nabla_i e(z_{-i}^{k-1}, u_i^{k-1})\|^2$$

$$\underset{(a)}{\leq} 8\eta^2 L_{max}^2\|u^{k+1} - u^k\|^2 + 4\eta^2 L_{max}^2\|u^k - u^{k-1}\|^2 + 16\eta\mathcal{F},$$

where in (a), we utilize Lipschitz continuity. Then, we obtain

$$\|u^{k+1} - u^k\|^2 \leq \frac{4\eta^2 L_{max}^2}{1 - 8\eta^2 L_{max}^2}\|u^k - u^{k-1}\|^2 + \frac{16\eta\mathcal{F}}{1 - 8\eta^2 L_{max}^2}, \qquad (16)$$

Considering $w \triangleq \frac{4\eta^2 L_{max}^2}{1 - 8\eta^2 L_{max}^2}$, we have

$$\|u^{k+1} - u^k\|^2 \leq w\|u^k - u^{k-1}\|^2 + \frac{16\eta\mathcal{F}}{1 - 8\eta^2 L_{max}^2}$$

$$= w^k\|u^1 - u^0\|^2 + \sum_{t=0}^{k-1}w^t\frac{16\eta\mathcal{F}}{1 - 8\eta^2 L_{max}^2}$$

$$\underset{(a)}{\leq} \frac{1 - w^t}{1 - w}\frac{16\eta\mathcal{F}}{1 - 8\eta^2 L_{max}^2} \leq \frac{1}{1 - w}\frac{16\eta\mathcal{F}}{1 - 8\eta^2 L_{max}^2} < 1.14 * 16\eta\mathcal{F} < 18.2\eta\mathcal{F},$$

where (a) is true because we have $\|u^1 - u^0\|^2 \le 16\eta\mathcal{F}$ since $k < K$ and Eq. (15), and we use $\eta \le c'_{max}/L_{max}$ where $c'_{max} = 0.1$ such that $w \approx 0.0435 < 1$. Then we obtain

$$\|u^{k+1} - u^k\| \le 4.3\sqrt{\eta\mathcal{F}} \le \frac{4.3\eta\mathcal{G}}{\mathcal{K}}. \tag{17}$$

Then, we get the upper bound of the sum of $\|u^{k+1} - u^k\|$, $\forall k < K$ as follows,

$$\sum_{t=1}^{k+1} \|u^t - u^{t-1}\| \le \sqrt{k\sum_{t=1}^{k+1} \|u^t - u^{t-1}\|^2} \le K\frac{4.3\eta\mathcal{G}}{\mathcal{K}} \le \hat{c}\mathcal{T}\frac{4.3\eta\mathcal{G}}{\mathcal{K}} \le 4.3\hat{c}\mathcal{S}, \tag{18}$$

which implies

$$\|u^{k+1}\| \underset{(a)}{\le} \sum_{t=1}^{k+1} \|u^t - u^{t-1}\| + \|u^0\| \le 4.3\hat{c}\mathcal{S}, \tag{19}$$

where in (a) we use the triangle inequality and $u^0 = 0$. Due to the following

$$\|u^{k+1} - \tilde{v}^k\| = \|u^{k+1} - u^0 + u^0 - \tilde{v}^k\| \le \|u^{k+1} - u^0\| + \|u^0 - \tilde{v}^k\| \le 4.3\hat{c}\mathcal{S} + \mathcal{S}/(2\mathcal{K}log(\frac{d\mathcal{K}}{\delta})), \tag{20}$$

we have $\|u^{k+1} - \tilde{v}^k\| \le 5\hat{c}\mathcal{S}$ since $\hat{c} \ge 2$. Therefore, we know that there exits $c^1_{max} = min\{c_{max}, c'_{max}\}$ such that $\|u^k - \tilde{v}^k\| \le 5\hat{c}\mathcal{S}$, $\forall k < K$ when $\eta \le c^1_{max}/L_{max}$. The proof is finished.

For **step 2**, we provide the following proof.

**Proof.** Given $\tilde{v}^k$ as an $\epsilon$-second order stationary point.

$$\mathcal{H}_u \triangleq \begin{bmatrix} \nabla^2_{11}e(\tilde{v}^k) & \nabla^2_{12}e(\tilde{v}^k) \\ 0 & \nabla^2_{22}e(\tilde{v}^k) \end{bmatrix} \quad \mathcal{H}_l \triangleq \begin{bmatrix} 0 & 0 \\ \nabla^2_{21}e(\tilde{v}^k) & 0 \end{bmatrix}, \tag{21}$$

Conclusion 1 is showing that the maximum eigenvalue of $M^{-1}G$ is greater than 1 so that we can project iterates $v^k$ to two subspaces. The first is spanned by the eigenvector of $M^{-1}G$ whose eigenvalue is greater than 1 and the other subspace is spanned by the remaining eigenvectors. Considering $det(M) = 1$, $det(M^{-1}G - \lambda I) = det(G - \lambda M)$, where $\lambda$ is the eigenvalue. The determinant of $G - \lambda M$ is shown as

$$det[G - \lambda M] = det[I - \eta\mathcal{H}_u - \lambda(I + \eta\mathcal{H}_l)]$$
$$= det\begin{bmatrix} (1-\lambda)I - \eta\nabla^2_{11}e(\tilde{v}^k) & -\eta\nabla^2_{12}e(\tilde{v}^k) \\ -\lambda\eta\nabla^2_{21}e(\tilde{v}^k) & (1-\lambda)I - \eta\nabla^2_{22}e(\tilde{v}^k) \end{bmatrix}.$$

To simplify, $Q(\lambda) \triangleq \begin{bmatrix} (1-\lambda)I - \eta\nabla^2_{11}e(\tilde{v}^k) & -\eta\nabla^2_{12}e(\tilde{v}^k) \\ -\lambda\eta\nabla^2_{21}e(\tilde{v}^k) & (1-\lambda)I - \eta\nabla^2_{22}e(\tilde{v}^k) \end{bmatrix}$. To illustrate the establishment of $\lambda_{max}(M^{-1}G) > 1$, we need to prove two condition: 1) All eigenvalues of $Q(\lambda)$ are real. 2) $det(Q(\lambda)) = 0, \exists\lambda > 1$.

Given a $\delta > 0$,

$$Q(1 + \delta) = -(\eta\mathcal{H} + \delta(I + \eta\mathcal{H}_l)) \triangleq -F(\delta) \tag{22}$$

$F(\delta)$ can be expended to

$$F(\delta) = \delta I + \eta\begin{bmatrix} \nabla^2_{11}e(\tilde{v}^k) & \nabla^2_{12}e(\tilde{v}^k) \\ (1+\delta)\nabla^2_{21}e(\tilde{v}^k) & \nabla^2_{22}e(\tilde{v}^k) \end{bmatrix}$$
$$= \begin{bmatrix} I & \\ & \sqrt{1+\delta} \end{bmatrix}\begin{bmatrix} \delta I + \eta\nabla^2_{11}e(\tilde{v}^k) & \eta\sqrt{1+\delta}\nabla^2_{12}e(\tilde{v}^k) \\ \eta\sqrt{1+\delta}\nabla^2_{21}e(\tilde{v}^k) & \delta I + \eta\nabla^2_{22}e(\tilde{v}^k) \end{bmatrix}\begin{bmatrix} I & \\ & \frac{1}{\sqrt{1+\delta}} \end{bmatrix}.$$

To simplify, $O(\delta) \triangleq \begin{bmatrix} \delta I + \eta\nabla^2_{11}e(\tilde{v}^k) & \eta\sqrt{1+\delta}\nabla^2_{12}e(\tilde{v}^k) \\ \eta\sqrt{1+\delta}\nabla^2_{21}e(\tilde{v}^k) & \delta I + \eta\nabla^2_{22}e(\tilde{v}^k) \end{bmatrix}$. Since $F(\delta)$ is similar to $O(\delta)$, $F(\delta)$ has the same eigenvalues of $O(\delta)$. Since $\mathcal{H}$ and $O(\delta)$ are diagonalizable (normal matrices),

then according to [7, 17], the result of quantifying the difference of the eigenvalues of the two normal matrices

$$\max_{1 \le i \le d} |\lambda_i(\eta\mathcal{H}) - \lambda_i(O(\delta))| \le \|\eta\mathcal{H} - O(\delta)\| \tag{23}$$

where $\lambda_i(\mathcal{H})$ and $\lambda_i(O(\delta))$ denote the i-th eigenvalue of $\mathcal{H}$ and $O(\delta)$, which are listed in a decreasing order. We expand Eq. (23) as

$$
\begin{aligned}
&\|\eta\mathcal{H} - O(\delta)\| \\
&= \|\delta I + \begin{bmatrix} 0 & (\sqrt{1+\delta} - 1)\eta \bigtriangledown_{12}^2 e(\tilde{v}^k) \\ (\sqrt{1+\delta} - 1)\eta \bigtriangledown_{21}^2 e(\tilde{v}^k) & 0 \end{bmatrix}\| \\
&\le \delta + (\sqrt{1+\delta} - 1)\eta\|\mathcal{H}\| + (\sqrt{1+\delta} - 1)\eta \begin{bmatrix} \bigtriangledown_{11}^2 e(\tilde{v}^k) & 0 \\ 0 & \bigtriangledown_{22}^2 e(\tilde{v}^k) \end{bmatrix} \\
&\le \delta + (\sqrt{1+\delta} - 1)(\frac{C}{C_{max}} + 1)
\end{aligned}
\tag{24}
$$

When $\eta \le c_{max}/C_{max}$, Eq. (24) is true according to $\|\mathcal{H}\| \le C$ and $\|\mathcal{H}_d\| \le C_{max}$. Specifically, when $\delta = 0$, matrix $O(\delta)$ is reduced to $\eta\mathcal{H}$. If $\eta = 1/C$, $\|\eta\mathcal{H} - O(\delta)\| \le \delta + 2(\sqrt{1+\delta} - 1)$.

Since the minimum eigenvalue of $\eta\mathcal{H}$ is $-\eta\gamma$ and the maximum difference of the eigenvalues between $\eta\mathcal{H}$ and $O(\delta)$ is upper bounded by Eq. (24). We need to set a $\delta$ to satisfy $\delta + (\sqrt{1+\delta} - 1)(C/C_{max} + 1) \le \eta\gamma$. It means that $O(\delta)$ has a negative eigenvalue if $\delta$ is sufficiently small.

Therefore, if $\delta^* = \eta\gamma/(1 + C/C_{max})$, $O(\delta^*)$ has a negative eigenvalue which is less than $-\eta\gamma + \delta^* = -\eta\gamma/(1 + C_{max}/L)$. In the following, we need to check that $\delta + (\sqrt{1+\delta} - 1)(C/C_{max} + 1) \le \eta\gamma$ holds when $\delta^* = \eta\gamma/(1 + C/C_{max})$.

Since $C/C_{max} \ge 1$, we get $\eta\gamma/(1 + C/C_{max}) \le \eta\gamma/2$. Accordingly, we need to check

$$
\begin{aligned}
&(\sqrt{1+\delta} - 1)(\frac{C}{C_{max}} + 1) < \frac{\eta\gamma}{2} \\
&\Longrightarrow \sqrt{1+\delta}(\frac{C}{C_{max}} + 1) < \frac{\eta\gamma}{2} + (\frac{C}{C_{max}} + 1) \\
&\Longrightarrow (1+\delta)(\frac{C}{C_{max}} + 1)^2 < (\frac{\eta\gamma}{2} + (\frac{C}{C_{max}} + 1))^2
\end{aligned}
\tag{25}
$$

Then we take $\delta^*$ into Eq. (25)

$$
\begin{aligned}
(1 + \frac{\eta\gamma}{1 + \frac{C}{C_{max}}})(\frac{C}{C_{max}} + 1)^2 &\le (\frac{C}{C_{max}} + 1)^2 + (\frac{C}{C_{max}} + 1)^2\eta\gamma \\
&< (\frac{C}{C_{max}} + 1)^2 + (\frac{C}{C_{max}} + 1)^2\eta\gamma + \frac{\eta^2\gamma^2}{4}
\end{aligned}
\tag{26}
$$

Specifically, $(\frac{\eta\gamma}{2} + (\frac{C}{C_{max}} + 1))^2 = (\frac{C}{C_{max}} + 1)^2 + (\frac{C}{C_{max}} + 1)^2\eta\gamma + \frac{\eta^2\gamma^2}{4}$. Therefore, $Q(1 + \delta^*)$ has a negative eigenvalue. Similarly, when $\delta$ is large, $Q(1 + \delta^*)$ has a positive eigenvalue, since term $\delta^2 I$ dominates the spectrum of matrix $Q(1 + \delta)$ in Eq. (22). We can conclude there exists a largest $\tilde{\delta}$, making $Q(1 + \tilde{\delta})$ has a zero eigenvalue because the eigenvalue is continuous w.r.t $\delta$. $det(Q(1 + \tilde{\delta})) = 0, 1 + \tilde{\delta} > 1 + \delta^* > 1$. Therefore, there exists a largest real eigenvalue of $M^{-1}G$ greater than 1. The proof is finished.

## 2   1-vs-1 transformation

To achieve DLN reusability, we convert multi-category classification into binary classification using the 1-vs-1 approach. To illustrate this process, we consider a three-category problem. The classifier output, denoted as $f_\theta(x) = (p_0, p_1, p_2)$, shows the predictions for each category. We combine the prediction for the correct category along with predictions for each incorrect category. This process enables us to change the three-category classification into two binary classification tasks: $(p_0, p_1)$, and $(p_0, p_2)$. We use the sigmoid function to normalize the output of the student model.

For semantic segmentation, *e.g.*, VOC [2], each pixel is classified among 21 classes. The PSPNet [18] output dimension is $[1, 21, 380, 380]$, meaning each sample contains $380 \times 380$ pixels classified into one of the 21 classes. First, we adjust the output dimension to $[1, 380 \times 380, 21]$. Then, we use the 1-vs-1 method to convert the $380 \times 380$ tasks of 21 classes into $380 \times 380 \times 20$ binary classification tasks.

## 3 Coordinate-wise and preprocessing

The teacher model encounters two challenges. First, optimizing a fully connected LSTM with numerous parameters is unfeasible since it demands a large hidden state and an excessive amount of parameters. Second, the magnitudes of inputs and outputs may vary significantly, depending on the function class under optimization. Nonetheless, neural networks work best when handling inputs and outputs that are neither very small nor very large. We adopt the approach described in Andrychowicz et al. [1] to address these difficulties.

To tackle the first challenge, the teacher model executes coordinate-wise operations on the DLN parameters, reminiscent of Adam [10] and RMSProp [5]'s update policies. As a result, we can employ a concise model by specifying the teacher model only, while using shared parameters for multiple DLN components. Each DLN parameter is assigned separate activations to ensure distinct behavior for different coordinates. This configuration induces teacher model invariance to the DLN parameters sequence.

Andrychowicz et al. [1] tackled the second challenge by introducing a pre-processing technique that enables gradient scaling adjustment by separating its information in terms of magnitude and sign. The method explained in Eq (27), presents two distinct cases. The first is when the input $|x|$ is greater than or equal to $e^{-p}$, whereas the second is when it is less than $e^{-p}$. Here $e$ denotes the Euler number. To keep it consistent with our experiment settings, we set $p$ as a constant equal to 10.

$$x \rightarrow \begin{cases} (\dfrac{log(|x|)}{p}, sgn(x)), & if |x| \geq e^{-p}, \\ (-1, e^p x), & otherwise. \end{cases} \tag{27}$$

## 4 The loss functions of YOLO-v3

The loss functions of YOLO-v3 [15] are comprised of three parts: 1) bounding box prediction $\ell_{loc}$, 2) class prediction $\ell_{cls}$, and 3) confidence level of the bounding box containing objects $\ell_{conf}$. In experiments using COCO dataset [13], YOLO-v3 predicts 3 boxes at each scale, resulting in a tensor of $[416, 416, 3 * (4 + 1 + 80)]$ for the 4 bounding box offsets, 1 objectiveness prediction, and 80 class predictions.

$\ell_{loc}$ is calculated as following

$$\ell_{loc} = \sum_{i=0}^{V^2} \sum_{j=0}^{2} 1_{i,j}^{obj} (2 - q_{2,i} q_{3,i}) [(q_{0,i} - \tilde{q}_{0,i})^2 + (q_{1,i} - \tilde{q}_{1,i})^2 + (q_{2,i} - \tilde{q}_{2,i})^2 + (q_{3,i} - \tilde{q}_{3,i})^2], \tag{28}$$

where $V$ denotes the grid size, $1_{i,j}^{obj}$ denotes whether there is object contained in $(i, j)$ (if so, $1_{i,j}^{obj} = 1$, otherwise $1_{i,j}^{obj} = 0$). Here $q_0$, $q_1$, $q_2$ and $q_3$ represent the label of the 4 bounding box offsets, and $\tilde{q}_0$, $\tilde{q}_1$, $\tilde{q}_2$ and $\tilde{q}_3$ represent the prediction of 4 bounding box offsets.

$\ell_{cls}$ is calculated as following

$$\ell_{cls} = \sum_{i=0}^{V^2} \sum_{j=0}^{2} 1_{i,j}^{obj} BCELoss(p_i, \tilde{p}_i), \tag{29}$$

where $BCELoss$ stands for the binary cross entropy loss. $p$ denotes the class predictions, and $\tilde{p}$ denotes the labels.

Finally, $\ell_{conf}$ is calculated as following

$$\ell_{conf} = \sum_{i=0}^{V^2} \sum_{j=0}^{2} 1_{i,j}^{obj} (o_i - \tilde{o}_i)^2 + \sum_{i=0}^{V^2} \sum_{j=0}^{2} 1_{i,j}^{noobj} (o_i - \tilde{o}_i)^2, \tag{30}$$

Table 1: Results on datasets CIFAR-10 [11], CIFAR-100 [12] with noisy label for the classification task. All experiments are implemented with the same settings. The best results are highlighted in bold.

| Method | CIFAR-10 | | CIFAR-100 | |
|---|---|---|---|---|
| | p=20% | p=40% | p=20% | p=40% |
| Baseline | 76.83 | 70.77 | 50.86 | 43.01 |
| MentorNet [8] | 86.36 | 81.76 | 61.97 | 52.66 |
| Meta-Weight-Net [16] | 90.33 | 87.54 | 64.22 | 58.64 |
| L2R [4] | 91.05 | 88.71 | 66.08 | 60.51 |
| Ours | **92.11±0.27** | **89.39± 1.20** | **70.05± 0.23** | **61.27± 0.51** |

where $1_{i,j}^{noobj}$ indicates whether cell $(i, j)$ contains object or not (if so, $1_{i,j}^{obj} = 0$, otherwise $1_{i,j}^{obj} = 1$), and $o$ represents the objectiveness prediction.

## 5   More experiments of image classification

We utilize a straightforward data reweighting method in both clean-label and noisy-label classification tasks. We determine the appropriate weight for each training data using the student model's embedding space to calculate the importance of each training data. To evaluate the importance of training data, we consider the validation data as the student model's unseen data. The more uniform the distribution of validation data, the more critical the significance of training data. We measure the importance of each training data by how comparable they are to the corresponding validation class center. To maintain student learning consistency, we need to normalize the weights within a minibatch. For training data $x_i$, $i = \{1, \cdots, batchsize\}$, after calculating the weight $\tilde{w}_i$, we normalize it to $w_i = \tilde{w}_i / \sum_{i=1}^{batchsize} \tilde{w}_t$ to ensure that the sum of weights is always equal to 1.

### 5.1   Noisy-label classification

**Datasets.** For the Noisy-label classification, We use two datasets: CIFAR-10 [11], CIFAR-100 [12]. Cifar-10 and Cifar-100 contain 50000 training and 10000 testing images and with 10-class and 100-class separately. We modify their labels by randomly flipping them to two inherently similar classes with an equal probability, as per the protocol of Shu et al. [16]. The flipping is conducted with an independent probability of $p$ for each of all images. In our experiments, we adopt ResNet32 as the student model and varied $p$ between $20\%$ and $40\%$. Following the experimental settings of the L2R [4], both the validation and test sets are clean.

**Evaluation metrics.** In the classification, we use the accuracy on the testing set of each dataset [8, 16, 4].

**Baseline methods.** We employ several popular methods, including Cross Entropy loss (CE), the MentorNet [8], the Meta-Weight-Net [16], and the L2R [4].

**Implementation details.** For all our experiments, we utilize the standard stochastic gradient descent (SGD) to optimize student models with a learning rate of 0.1, while employing Adam with a learning rate of 0.001 for the teacher model. The learning rate of DLN is set to 0.001. The teacher model is trained for 10 epochs, with redividing the training and validation data after each epoch. The validation errors in each task are explicitly reported. For the teacher model, we employ a four-layer LSTM [6] with 64 neurons in the first three layers and 1 neuron in the last layer. We use 1-vs-1 to process the output of the student model in classification. To ensure a reliable evaluation, we conduct 5 random restarts and use the average results for comprehensive comparisons.

**Results.** For the DLN, we perform a five-layer fully connected network, which contains 40 neurons in each hidden layer and 1 neuron in the output layer, as the DLN. The activation function for each hidden layer is set to Leaky-ReLU. The validation error is computed by CE. The results are presented in Table 1. Our study has demonstrated that the combined strategy of data reweighing and dynamic loss function resulted in better outcomes than previous works such as MentorNet [8], MetaWeight-Net [16], and L2R [3]. More precisely, in CIFAR-10 and CIFAR-100 datasets, the L2T-DLN method outperformed L2R by $1.06\%$ and $3.97\%$ ($p = 20\%$) and $0.68\%$ and $0.76\%$ ($p = 40\%$) respectively.

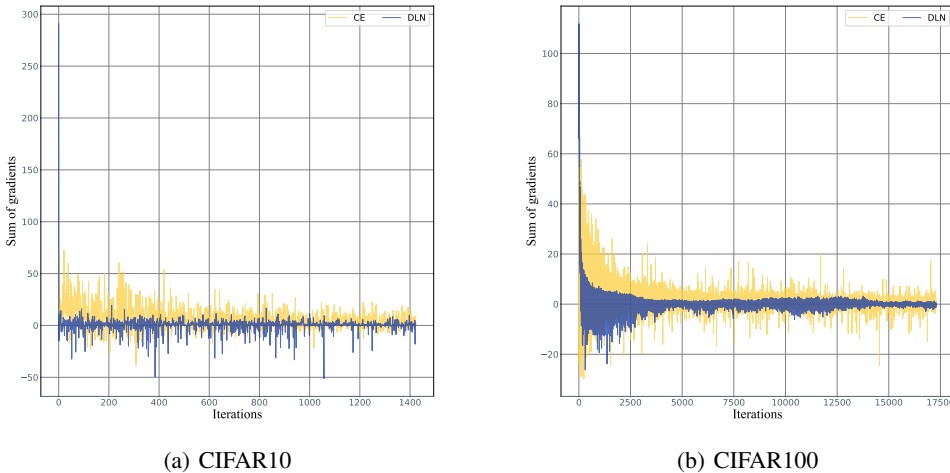

(a) CIFAR10            (b) CIFAR100

Figure 1: Illustration of the sum of gradients provided to the student model by L2T-DLN (blue) and CE (yellow). (a) and (b) indicate that: 1) DLN initially generates a steep gradient; 2) with the training of the teacher model and student model, the gradient of the DLN becomes smoother than CE; 3) the teacher model and student model in CIFAR-100 converge more slowly than that in the CIFAR-10 task. We conclude that our L2T-DLN is smoother and more robust for student learning than CE.

To compare L2T-DLN and CE, we illustrate the sum of gradients provided to the student model at each iteration when $p = 20\%$ using Figure 1. The results indicate that DLN initially generates a sharp gradient, but during the training of both the teacher and student models, the gradient becomes smoother than CE. Furthermore, Figure 1 shows that the teacher and student models in CIFAR-100 tend to converge more slowly than those in CIFAR-10. In summary, our L2T-DLN approach is smoother and has more robust learning for student models compared to CE.

## 5.2 Ablations

In this subsection, we conduct ablation studies on CIFAR-10 [11] using ResNet8 to analyze the impact of different Similarity metrics, the influence of the LSTM teacher and data reweighting, and the influence of wrong learning rate settings for students.

**Similarity metrics.** Since the data reweighting requires calculating similarity, we discuss the influence of different similarity metrics. We compare the performance of four similarity metrics (cosine, dot product, Minkowski distance, and signal-to-noise ratio(SNR)).

The formula of cosine is shown as:

$$cos(\vec{a}, \vec{b}) = \frac{a \cdot b}{\| a \| \| b \|} = \frac{\sum_{i=1}^{n} a_i \times b_i}{\sqrt{\sum_{i=1}^{n}(a_i)^2} \times \sqrt{\sum_{i=1}^{n}(b_i)^2}} \tag{31}$$

The formula of dot product is shown as:

$$dot(\vec{a}, \vec{b}) = \sum_{i=1}^{n} a_i \times b_i \tag{32}$$

The formula of Minkowski distance is shown as:

$$L_p(\vec{a}, \vec{b}) = (\sum_{i=1}^{n} |a_i - b_i|^p)^{\frac{1}{p}}, p \geq 1 \tag{33}$$

Specifically, We set $p = 2$.

The formula of SNR is shown as:

$$SNR(\vec{a}, \vec{b}) = 10log(PS/PN), PS = \sigma(\vec{a}), PN = \sigma(\vec{a} - \vec{b}) \tag{34}$$

Table 2: Results on different metrics including cosine, dot product, Minkowski distance, and signal-to-noise ratio(SNR) to show the impact of metrics.

| similarity | cosine | Dot Product | $L_p$ | SNR |
|---|---|---|---|---|
| accuracy | 89.20 | 89.27 | 89.32 | 90.70 |

$\sigma(\vec{a})$ means the variance of $\vec{a}$. Table 2 summarizes the results of the ablation study. Cosine, dot product, and Minkowski distance all impose strict constraints on embedding space for the student model. These metrics do not account for the distribution of data points, making them deterministic in nature. In contrast, SNR is a soft constraint that considers only the amount of information present in each data point and disregards its distribution. Based on the features of similar measures and the outcomes of the ablation study, we infer that SNR can aid the student model in achieving better performance.

**LSTM and data reweighting.** In this section, we examine the effect of data reweighting (DR) on clean-label CIFAR-10 tasks using ResNet8. We initially optimized the student model with LDN as a loss function utilizing Adam to adjust its parameters. We then analyzed the effect of each element by conducting DLN+LSTM, DLN+DR, and DLN+DR+LSTM. The results are summarized in Table 3. Our analysis revealed that the performance of DLN improved by 0.23 when DR was used and 0.48 when LSTM was incorporated. Moreover, the LSTM teacher's influence was found to be more significant.

Table 3: Results on different learning elements to show the impact of the LSTM teacher and the data reweighting.

| DLN | DR | LSTM | ResNet8 |
|---|---|---|---|
| ✓ | | | 90.17 |
| ✓ | | ✓ | 90.65 |
| ✓ | ✓ | | 90.40 |
| ✓ | ✓ | ✓ | 90.70 |

**Learning Rate (LR) settings.** In this ablation study, we examine the impact of 'wrong' learning rates on the performance of L2T with dynamic loss. We employ varying learning rates (0.2, 0.3, 0.4, and 0.5) for the student model, subsequently analyzing the disparities in performance between our DLN and other L2T baseline methods, *e.g.*, SLF, within the context of the CIFAR10-ResNet8 task. The results are presented in Table 4. In comparison to SLF, our proposed method has achieved a relatively higher accuracy even with a 'wrong' learning rate. The above comparison indicates that our method outperforms the previous state-of-the-art SLF under the wrong learning rate (0.4/0.5) setting. Compared with SLF, our method is less sensitive to the 'wrong' learning rate.

Table 4: Results on wrong learning rates ranging from 0.2 to 0.5 to show the impact of different learning rates.

| LR | 0.2 | 0.3 | 0.4 | 0.5 |
|---|---|---|---|---|
| SLF | 89.2 | 87.5 | 66.0 | 13.5 |
| ours | 89.7 | 88.1 | 70.4 | 49.0 |

# 6   Inferences of semantic segmentation

The semantic segmentation results are presented in Figure 2. Compared to the origin PSPNet [18], the advantages of our L2T-DLN are summarised as follows: 1) accurately segment subtle foreground; 2) focusing on the reasoning of the occluded foreground. For instance, in the second line, our model segments the horse's front legs more accurately than PSPNet and focus on the unoccluded part of the

hind legs. Similarly, in the fourth line, our model clearly shows that one leg of the cyclist is occluded by the bicycle, whereas the original PSPNet misclassifies it as the cyclist.

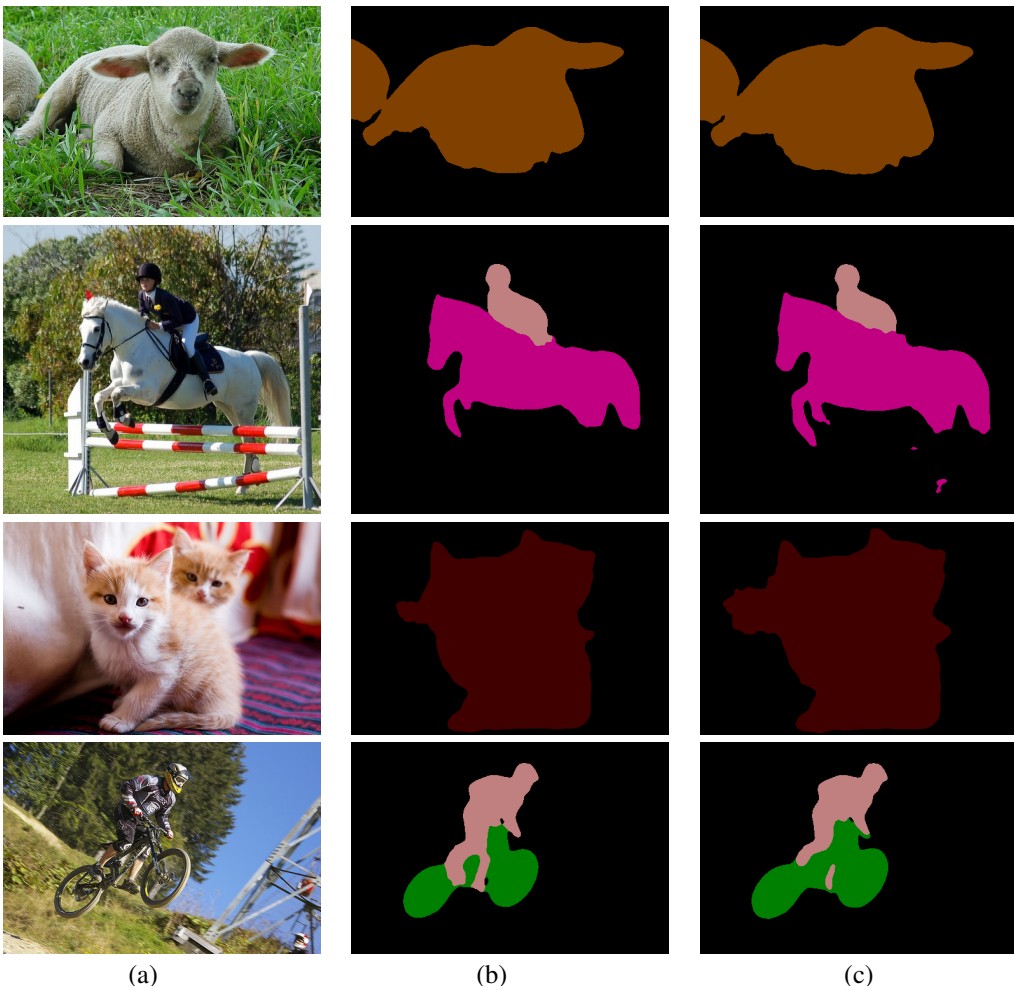

     (a)                        (b)                       (c)

Figure 2: Illustration of results of the semantic segmentation. (a) are original images; (b) are results of original PSPNet; (c) are results of PSPNet-ours. As shown in the first and third lines, when the foreground is uncovered, our segmentation results are similar to the original PSPNet. The second and fourth lines show the advantages of our L2T-DLN as follows: 1) accurately segment subtle foreground; 2) focus on the reasoning of the occluded foreground. In the second line, we segment the horse's front legs more accurately than PSPNet and focus on the unoccluded part of the hind legs. Similarly, in the fourth line, our model clearly shows that one leg of the cyclist is occluded by the bicycle, whereas the original PSPNet misclassifies it as the cyclist.