# OpenReview forum: "L2T-DLN: Learning to Teach with Dynamic Loss Network"
_NeurIPS.cc/2023/Conference — NeurIPS 2023 poster_

### Official Review · Reviewer_HMfR · 2023-06-20

**Soundness:** 2 fair
**Presentation:** 3 good
**Contribution:** 2 fair
**Rating:** 5
**Confidence:** 3

**Summary:**

The paper introduces the concept of teaching in machine learning and proposes a framework called L2T-DLN (Learning to Teach with Dynamic Loss Network). The framework aims to address the limitations of existing approaches by incorporating the temporal nature of loss function adjustment and utilizing the states of the loss function. The authors formulate the loss adjustment as a temporal task using a teacher model with memory units and propose a dynamic loss network to enhance the interactions between the teacher and student model. Extensive experiments demonstrate the effectiveness of the approach on various real-world tasks.

**Strengths:**

1. The paper presents a novel framework, L2T-DLN, which integrates the concept of teaching with dynamic loss functions. This approach stands out by considering the temporal nature of loss function adjustment and utilizing the states of the loss function, setting it apart from existing works.
2. The authors provide a thorough convergence analysis of the proposed framework, demonstrating its effectiveness and its ability to achieve convergence.

**Weaknesses:**

1. Unclear Motivation: The paper lacks a thorough discussion of the benefits of introducing the gradients concerning loss functions. It would be beneficial to include a clearer motivation.
2. Marginal Improvement: The reported improvements of 0.4% lower than ALA on ImageNet for classification and only 0.3% improvement compared to the baseline on mIoU on the VOC dataset for segmentation are relatively small.


**Questions:**

1. Further clarification is needed in the paper on why it is necessary to consider the gradients concerning loss functions as a core idea. Besides experimental results, could there be theoretical analysis supporting this aspect?
2. The authors introduce a Dynamic Loss Network instead of using a dynamic loss function. It would be helpful to explain if this choice leads to a larger number of parameters and potential unfairness in comparisons. Please clarify the impact of parameter quantity in this context.

**Limitations:**

The implementation of this approach necessitates substantial computational resources for calculating high-order derivatives.

---

> ### Author Rebuttal · Authors · 2023-08-09
>
> Thank you for your valuable comments.
>
> Weakness:
>
> 1. Previous studies involve directly supplying certain states of the student model to a teacher for dynamic loss adjustment (kindly refer to Lines 28-30, main paper). This direct provision of states without integration hinders L2T convergence, as the teacher model necessitates further learning to integrate these state-based insights effectively. In contrast, the gradient concerning DLN achieves holistic information integration throughout the learning process (kindly refer to Lines 130-136, main paper), facilitated by prior knowledge (chain rule). Employing the gradient concerning the loss allows the teacher model to concentrate on capturing and preserving crucial information from gradients, negating the need for supplementary handling of dispersed states.
>
> 2. The performance of ALA (reference [15], main paper) benefits from two aspects:
>
>     (1) The length of a student learning stage ALA set (200) is much larger than ours (25). Note our ablation showed that the size of the length is positively correlated with the test accuracy and computational consumption (Lines 276-283, main paper);
>
>     (2) ALA employed a multi-student training strategy to achieve stronger performance with an extra 20%-50% cost over other methods [A].
>
>     Therefore, considering its computational resources, our improvement is not marginal compared to ALA.
>
> Questions:
>
> 1. (1) For the benefits of the gradient concerning the loss, please refer to Weakness 1;
>
>     (2) Please kindly refer to Lines 130-136, main paper, for theoretical analysis. The gradient concerning the DLN involves the information of both training and validation data. As these pieces of information depend on each other, they are integrated into the temporal changes of the student. The gradient concerning DLN plays a crucial role in fusing these pieces of information. Therefore, compared to the state of the student, the gradient is the integration of information throughout the learning process, which provides more information to promote deep interaction between DLN, teacher, and student.
>
> 2. (1) At the testing stages, the number of parameters of the student model is the same in ours and other baselines. The comparisons are fair. Note, in the L2T framework, the teacher and the dynamic loss network only affect the optimization of the student model during the training phase, while the student model operates independently during the testing phase;
>
>     (2) The number of parameters in DLN is 5000. Compared to commonly used models, e.g., the parameter of ResNet8 is 1235274, 5000 number of parameters is relatively small;
>
>     In response to the reviewer's concerns, we construct an experiment. We added 5000 parameters to the teacher model of baseline 'Stochastic Loss Function' (SLF, reference [22], main paper). The performance of SLF on CIFAR-10+ResNet8 with a testing accuracy of 89.97%, while our L2T-DLN still maintains a performance advantage (90.65%).
>
> [A] Huang, C., Zhai, S., Talbott, W., Bautista, M. A., Sun, S. Y., Guestrin, C., & Susskind, J. Addressing the Loss-Metric Mismatch with Adaptive Loss Alignment Supplementary Material.

---

> > ### Comment · Reviewer_HMfR · 2023-08-14
> >
> > The response solves my concerns. I suggest incorporating the comparison of "the length of a student learning stage" into Table 1 to better highlight the superiority of DLN. Thus, I improve my final rating from 4 to 5.

---

> > > ### Author Response · Authors · 2023-08-14
> > > **Thanks to reviewer HMfR**
> > >
> > > Many thanks for all the helpful comments and positive assessments.
> > >
> > > We really appreciate reviewer HMfR for upgrading the score. Thank you again for your valuable comments and time efforts. We will incorporate the comparison of "the length of a student learning stage" into Table 1 to better highlight the superiority of DLN.

---

### Official Review · Reviewer_79Na · 2023-07-05

**Soundness:** 4 excellent
**Presentation:** 3 good
**Contribution:** 4 excellent
**Rating:** 7
**Confidence:** 4

**Summary:**

This paper introduces an improvement towards the learning-to-teach framework. Compared with the previous works, the authors made an innovation that a dynamic loss network is added with LSTM acting as teacher model to enhance the temporal memorization. A three step optimization procedure is also proposed with an convergence analysis theoretically. Last extensive experiments are conducted on various computer vision tasks that demonstrated the effectiveness of the proposed approach.

**Strengths:**

1. The problem this paper studied is an important one yet insufficiently studied in the current literature, that is how to use a better teaching strategy in addition to merely conduct "learning". It's a very meaningful step forward of this paper that pushes forward the frontier of such a literature; Furthermore the design of dynamic loss network is a new perspective for this area as well, going beyond the dynamic loss function design.

2. The paper is technically very sound with enough insights and depth. I particularly appreciate the convergence analysis which is done first time for the machine teaching/learning-to-teach area, with a good depth leveraging the negative curvature.

**Weaknesses:**

As also stated in the paper, despite being theoretically sound and elegant, the proposed algorithm incurs much more computational overhead which will hinder its practical usefulness in real world applications. Particularly the compute of Hessian even three-order derivatives is a sign of its computational complexity. I agree with the author that this needs to be improved further in the future to enlarge the impact of this work.

**Questions:**

While the memory units of LSTM make sense in this scenario, how about use Transformer as another sequential model which may even enhance the temporal memorization compared with LSTM?

**Limitations:**

Stated in the paper as well as the above point, that the computational complexity is one concern while it seems feasible and justifiable to further solve this, as the future potential value of this direction is likely large.

---

> ### Author Rebuttal · Authors · 2023-08-09
>
> 1. Our target for the submission is to re-visit L2T and formulate the loss adjustment as a temporal task. We hope our discussions inspire other further work. Thank you for your valuable and attractive suggestions for using transformer architecture as a teacher. Using a transformer as the teacher model involves the simultaneous processing of sequential data across timestamps. It might meet challenges, e.g., the accumulation of historical gradients and pressure on storage resources. We will explore the suggestion in the future.
>
> 2. Thank you for your valuable comments and efforts. We will attempt the following strategies in our future work:
>
>     (1) Design of appropriate prior knowledge for enhanced fusion to curtail the need for extensive higher-order gradient computations;
>
>     (2) Employing finite element method paradigm by partitioning the parameters associated with the student, DLN, and teacher models into discrete finite elements;
>
>     (3) Integration of Multivariate Taylor Series Expansion to more effectively navigate complex computations while optimizing resource consumption.

---

### Official Review · Reviewer_9Vkw · 2023-07-08

**Soundness:** 2 fair
**Presentation:** 1 poor
**Contribution:** 1 poor
**Rating:** 3
**Confidence:** 3

**Summary:**

In this paper, the authors state that existing methods only employ a simple feedforward network as the teacher model, which limits the potential of L2T. This issue motivates authors to propose a network with a memory unit to enhance the temporal analyzing ability of the teacher in the learning to teach (L2T) tasks. This proposed new network is combined with a dynamic loss network to address the issue of the existing works above. The experiment shows the superiority of the proposed method.

**Strengths:**

(+) This paper theoretically analyzes the convergence of the proposed method.

(+) The proposed method shows the state-of-the-art performance compared with other methods.


**Weaknesses:**

(-) The novelty of this paper is not clear. In the title, abstract and Introduction, the authors repeatedly emphasize that they propose an L2T framework with a Dynamic Loss Network (L2T-DLN). However, these emphases can cause misleading since Dynamic Loss has already been introduced in previous work, as stated by the authors themselves in Lines 2, 23, and the caption of Figure 1. Consequently, the application of Dynamic Loss in L2T cannot be considered as the novelty or the main contribution of the paper. The continued highlighting of DLN in these sections adds to the confusion and may hinder the understanding of the paper. In my understanding, the primary contribution of this paper is a proposed network with a memory unit to enhance the temporal analyzing ability of the teacher in the learning to teach (L2T) tasks. By emphasizing this aspect, rather than the Dynamic Loss network, the paper will become more coherent and easier to comprehend.

(-) The authors have not adequately demonstrated the effectiveness of their proposed method in both theory and practice. In the theoretical analysis, they solely focus on the convergence of the proposed L2T-DLN and fail to illustrate how the proposed network with a memory unit for the teacher can benefit the training of L2T task. Additionally, the experiments show the better performance of the proposed method compared with other SOTA methods. However, based on section 5.1, this better performance is based on many combinations of loss functions and tricks. It remains unclear whether the proposed network with a memory unit for the teacher contributes to the enhanced performance observed in the L2T task.

(-) The paper lacks sufficient ablation studies to support its claims. In the ablation study section, the authors include some experiments related to selecting the best learning parameters for their proposed method; these parameters may primarily influence performance rather than directly relating to the main contributions of the proposed method. Therefore, it is crucial for the authors to conduct a series of ablation studies to demonstrate how the proposed network with a memory unit for the teacher can benefit the training of L2T tasks.



**Questions:**

See weakness.

**Limitations:**

No limitations

---

> ### Author Rebuttal · Authors · 2023-08-09
>
> Thank you for your valuable comments.
>
> Weaknesses:
>
> 1. The novelty of our approach is introduced in Lines 8-13 and Lines 50-55, main paper, which are:
>
>     (1) design a teaching strategy based on the gradient concerning DLN;
>
>     (2) use LSTM as the teacher model to update the DLN with the temporal information;
>
>     (3) provide a convergence analysis of the approach.
>
>     Using a network with a memory unit to enhance the temporal analyzing ability of the teacher is one of our contributions.
>
>     The introduction of background knowledge, i.e., existing studies of L2T with dynamic loss, is necessary. Lines 2, 23, and Figure 1 (main paper), aim to help readers understand the difference between L2T-DLN and existing works about L2T with dynamic loss (DL). By realizing the differences, the readers will not be misled that the usage of DL is our contribution.
>
> 2. Please kindly refer to Section 3 and Section 5 for theoretical analysis and effectiveness demonstration of our approach. In practice, we evaluate three downstream tasks, i.e., image classification, object detection, and semantic segmentation. Experiments on three downstream tasks (refer to Section 5.2, main paper) and ablation study (refer to Lines 284-291, main paper) demonstrate that:
>
>     (1) L2T-DLN has shown state-of-the-art performance compared with other methods on different downstream tasks;
>
>     (2) The ablation study has drawn two conclusions: the first is that algorithms that can use historical information perform better than those that cannot, and the second is that the adaptability to capture and maintain short- and long-term dependencies can further enhance the loss function teaching, compared to handcrafted method.
>
> 3. Please kindly refer to Section 5.3 in our main paper and Section 5.2 in the supplementary material for the ablation study. The ablation study of w/o memory unit is shown in Lines 284-291, main paper, and Lines 215-221, supplementary material.

---

### Official Review · Reviewer_wSe9 · 2023-07-10

**Soundness:** 3 good
**Presentation:** 2 fair
**Contribution:** 2 fair
**Rating:** 5
**Confidence:** 3

**Summary:**

This paper presents a three-stage framework that dynamically adjusts the learning process of student model (i.e., target model). The loss value is calculated via the proposed Dynamic Loss Network (DLN), parameterized as neural network. And the DLN is updated by the teacher network implemented as LSTM. The authors also provide a convergence analysis. The model is evaluated on an array of tasks including image classification, object detection and semantic segmentation.

**Strengths:**

1. Compared to previous works, L2T-DLN leverages the state of loss function and the temporal information of student learning experience  to update the teacher.
2. Experiments on various application including image classification, object detection and semantic segmentation showcase the effectiveness.
3. The provided convergence analysis is helpful to understand the framework.

**Weaknesses:**

1. The proposed DLN is not unified. For image classification, it produces the loss value. For object detection, it generates the loss weights.
2. Some technical details of the DLN are missed, as will discussed in next session.
3. To access the temporal information of student learning, L2T-DLN would store the historical models and consume lots of memory.
4. Since the DLN adjusts the loss weights of the objectives of YOLO-v3, it’s reminiscent of multi-task learning. In this regard, the authors should discuss the relationship with MTL.

**Questions:**

1. The output layer (e.g., activation function) of the DLN is not clearly described.
2. It would be better to understand the DLN if the loss value/weights can be visualized.
3. Figure 1 in the supplementary material indicates the DLN (blue line) would produces lower gradients applied on the student model. Is this because the baseline model uses a large learning rate (0.1 line 219)?
4. The claimed effectiveness would be more convincing if an ablation study of learning rates is provided.
5. Regarding the initial gradients produced by the DLN, which greatly exceeds the baseline, it may be related to the network initialization of the DLN. Since this detail is missing, I would suggest the author clarifying the initialization.
6. For the DLN of object detection, is it initialized as identical mapping? If not, it’s helpful to investigate this manner.
7. What’s the training epoch for baseline model and the L2T-DLN for image classification?

**Limitations:**

1. Some details (e.g., initialization and activation of the output layer) of the proposed DLN are missing.
2. The used learning rate for image classification may not be optimal, an ablation study would help understand the framework.
3. It would be helpful to visualize the output of DLN.
4. The framework may consume lots of GPU memory.

---

> ### Author Rebuttal · Authors · 2023-08-09
>
> Thank you for your valuable comments.
>
> Weaknesses:
>
> 1. Objective detection involves two sub-tasks, i.e., regression and classification. It is challenging to handle two tasks simultaneously with one dynamic loss. An alternative way commonly used in existing dynamic loss-based works is to dynamically combine the objective functions of different tasks. Therefore, we follow their setting to perform our L2T-DLN.
>
> 2. We provide point-to-point responses in the next sessions.
>
> 3. Our memory unit stores the long- and short-term dependencies captured during the teaching process, disregarding the historical models. The size of the memory unit is fixed at 2.55MB and not consumes lots of memory.
>
> 4. L2T-DLN employs a dynamic loss set by a teacher to train a student model on a specific task, whereas MTL leverages potential correlations between diverse tasks to train a shared representation network. The L2T-DLN adjusts loss weights based on the temporal relationship between student performance and losses, for the same task, instead of using inter-task correlations as in MTL.
>
> Questions:
>
> 1. Our DLN is a fully connected network (kindly refer to Lines 233-235, Lines 246-247, and Line 257, main paper, and Lines 184-186, supplementary material), as well as the output layer. Following existing works, the output layer is not processed by any activation function.
>
> 2. We visualize the loss value of DLN on MNIST and CIFAR-10 separately in Figure 1 and 2, rebuttal. The DLN is initialized with the kaiming normal initialization with LeakyReLU activations.
>
> 3. No. Both our L2T-DLN and baseline follow the same learning rate settings of related work (references [3, 7, 15], supplementary material) about noise-label classification.
>
> 4. We evaluate the student model (ResNet8) concerning different learning rates on CIFAR-10. The learning rates were set to 1, 0.5, 0.1, 0.05, 0.01, 0.005, and 0.001, corresponding to accuracies of 18.8%, 49.0%, 90.7%, 87.6%, 88.1%, 87.3%, and 84.2%, respectively. Therefore, we set the learning rate as 0.1 in our experiment.
>
> 5. The DLN experiences random initialization through Kaiming normal initialization coupled with LeakyReLU activations for classification and segmentation tasks. For the objective detection task, all parameters of the DLN are initialized to a value of 1. There is no significant correlation between the substantial gradient and initialization of DLN during the initial training phases. In fact, in the early training stages, the significant DLN gradient prompts substantial changes in the student model, thereby broadening L2T's exploratory reach within the solution space. This expanded exploration facilitates the investigation of numerous local optimal solutions, enabling the selection of the most favorable among them.
>
> 6. Yes, it is initialized as identical mapping.
>
> 7. The baseline is trained with 200 epochs. In our approach, we set the teacher learning phase for 10 epochs (Line 222, main paper).

---

> > ### Comment · Reviewer_wSe9 · 2023-08-17
> >
> > Dear authors,
> >
> > Thanks for the response. This response solves most of my concerns.
> >
> > I still have some doubts about the used learning rate. The paper claims that it can learn to teach (i.e. optimize) the student model by the produced gradient or loss weights. I wonder if the proposed method is able to **correct** the optimization process if a **wrong** learning rate is used. As your response said, using a learning rate of 0.5 will result in a poor student baseline. Then, your method should **lower** the learning rate (i.e., gradients of low magnitude) to help the student improve the performance.
> >
> > Kind regards,

---

> > > ### Author Response · Authors · 2023-08-19
> > > **Thanks to reviewer wSe9**
> > >
> > > Thank you for your valuable and constructive comments, we appreciate this opportunity to respond to your comments and address your concerns.
> > > 1) Compared with other L2T baselines, e.g., 'Stochastic Loss Function (SLF)', our proposed method has achieved a relatively higher accuracy even with a `wrong` learning rate. Specifically, with dynamic loss and set the learning rate to 0.2, 0.3, 0.4, and 0.5 in the CIFAR-10-ResNet8 task, SLF archives 89.2, 87.5, 66.0, and 13.5 in accuracy, while our L2T-DLN achieve 89.7, 88.1, 70.4, and 49.0. The above comparison indicates that our method outperforms the previous state-of-the-art SLF under the wrong learning rate (0.4/0.5) setting. Compared with SLF, our method is less sensitive to the wrong learning rate.
> > >
> > > 2) To our knowledge, all L2T methods require setting a proper learning rate for optimization. For example, SLF and L2T-DLF set their learning rate as 0.1. The reason is that a large learning rate will lead to drastic changes in the student model and, therefore, beyond the teaching capability of the teacher model.

---

> > > > ### Comment · Reviewer_wSe9 · 2023-08-19
> > > >
> > > > Dear authors,
> > > >
> > > > Thanks for the clarification. I think this experiment highlights the novelty of your work. I am quite confident to improve my rating.
> > > >
> > > > I hope the authors will include this clarification in their final manuscript.
> > > >
> > > > Kind regards,

---

> > > > > ### Author Response · Authors · 2023-08-19
> > > > >
> > > > > Thank you very much for your suggestions. We are grateful for your support and will include the clarification in our final manuscript.

---

### Author Rebuttal · Authors · 2023-08-09

We thank all reviewers' valuable comments and efforts. Our proposed L2T-DLN is technically very sound with enough insights and depth [79Na], provides a thorough convergence analysis [wSe9, HMfR], and outperforms state-of-the-art methods in various downstream tasks [9Vkw,wSe9]. We provide point-to-point responses in the Official Review system.

---

### Decision · Program_Chairs · 2023-09-21

**Decision:**

Accept (poster)

**Comment:**

This paper proposes a novel L2T-DLN to enhance the temporal analyzing ability of the teacher in the learning to teach (L2T) tasks. Reviewer wSe9, 79Na and HMfR acknowledged the method novelty and experiments. After rebuttals, reviewer wSe9 think his/her concerns have been addressed. The area chair checks that authors also resolved well the problems of reviewer 79Na and HMfR. Reviewer 9Vkw gives negative comments. However the AC found the reviewer did not carefully read the paper and missed most of contents. The area chair thus decides to accept it.